# Riemannian Laplace approximations for Bayesian neural networks

**Federico Bergamin, Pablo Moreno-Muñoz, Søren Hauberg, Georgios Arvanitidis**
Section for Cognitive Systems, DTU Compute, Technical University of Denmark
{fedbe, pabmo, sohau, gear}@dtu.dk

## Abstract

Bayesian neural networks often approximate the weight-posterior with a Gaussian distribution. However, practical posteriors are often, even locally, highly non-Gaussian, and empirical performance deteriorates. We propose a simple parametric approximate posterior that adapts to the shape of the true posterior through a Riemannian metric that is determined by the log-posterior gradient. We develop a Riemannian Laplace approximation where samples naturally fall into weight-regions with low negative log-posterior. We show that these samples can be drawn by solving a system of ordinary differential equations, which can be done efficiently by leveraging the structure of the Riemannian metric and automatic differentiation. Empirically, we demonstrate that our approach consistently improves over the conventional Laplace approximation across tasks. We further show that, unlike the conventional Laplace approximation, our method is not overly sensitive to the choice of prior, which alleviates a practical pitfall of current approaches.

## 1 Introduction

*Bayesian deep learning* estimates the weight-posterior of a neural network given data, i.e. $p(\theta|\mathcal{D})$. Due to the generally high dimensions of the weight-space, the normalization of this posterior is intractable and approximate inference becomes a necessity. The most common parametric choice approximates the posterior with a Gaussian distribution, $p(\theta|\mathcal{D}) \approx q(\theta|\mathcal{D}) = \mathcal{N}(\theta|\mu, \Sigma)$, which is estimated *variationally* [Blundell et al., 2015], using *Laplace approximations* [MacKay, 1992] or with other techniques [Maddox et al., 2019]. Empirical evidence, however, suggests that the log-posterior is not locally concave [Sagun et al., 2016], indicating that the Gaussian approximation is overly crude. Indeed, this approximation is known to be brittle as the associated covariance is typically ill-

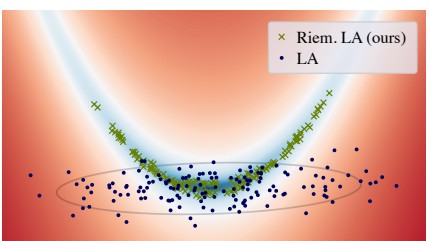

Figure 1: Our Riemannian Laplace approximation is a simple parametric distribution, which is shaped according to the local loss landscape through a Riemannian metric.

conditioned implying a suboptimal behavior [Daxberger et al., 2021a, Farquhar et al., 2020], and for this reason, alternative approaches have been proposed to fix this issue [Mackay, 1992]. Nonetheless, the Gaussian approximation is widely used due to the many benefits of parametric distributions, over e.g. *Monte Carlo sampling* [Neal, 1995] or *deep ensembles* [Lakshminarayanan et al., 2017].

**In this paper** we argue that the underlying issue is not with the Gaussian approximation, but rather with the weight-space over which the approximation is applied. We show that a Gaussian approximation can locally adapt to the loss by equipping the weight-space with a simple Riemannian metric and performing the approximation tangentially to the associated manifold. Practically, this ensures that samples from the Riemannian approximate posterior land in regions of weight-space yielding low training loss, which significantly improves over the usual Gaussian approximation. We obtain our

37th Conference on Neural Information Processing Systems (NeurIPS 2023).

Riemannian approximate posterior using a generalization of the Laplace approximation [MacKay, 1992] to general Riemannian manifolds. Sampling from this distribution requires solving a system of ordinary differential equations, which we show can be performed efficiently by leveraging the structure of the used Riemannian metric and automatic differentiation. Empirically, we demonstrate that this significantly improves upon conventional Laplace approximations across tasks.

## 2 Background

**Notation & assumptions.** We consider independent and identically distributed (i.i.d.) data $\mathcal{D} = \{\mathbf{x}_n, \mathbf{y}_n\}_{n=1}^N$, consisting of inputs $\mathbf{x} \in \mathbb{R}^D$ and outputs $\mathbf{y} \in \mathbb{R}^C$. To enable *probabilistic modeling*, we use a likelihood $p(\mathbf{y}|f_\theta(\mathbf{x}))$ which is either Gaussian (regression) or categorical (classification). This likelihood is parametrized by a deep neural network $f_\theta : \mathbb{R}^D \to \mathbb{R}^C$, where $\theta \in \mathbb{R}^K$ represent the weights for which we specify a Gaussian prior $p(\theta)$. The predictive distribution of a new test point $\mathbf{x}'$ equals $p(\mathbf{y}|\mathbf{x}') = \int p(\mathbf{y}|\mathbf{x}', \theta)p(\theta|\mathcal{D})\mathrm{d}\theta$ where $p(\theta|\mathcal{D})$ is the true weight-posterior given the data $\mathcal{D}$. To ensure tractability, this posterior is approximated. This paper focuses on the Laplace approximation, though the bulk of the methodology applies to other approximation techniques as well.

### 2.1 The Laplace approximation

The Laplace approximation (LA) is widely considered in *probabilistic* models for approximating intractable densities [Bishop, 2007]. The idea is to perform a second-order Taylor expansion of an unnormalized log-probability density, thereby yielding a Gaussian approximation. When considering inference of the true posterior $p(\theta|\mathcal{D})$, LA constructs an approximate posterior distribution $q_{\text{LA}}(\theta|\mathcal{D}) = \mathcal{N}(\theta|\theta_*, \Sigma)$ that is centered at the *maximum a-posteriori* (MAP) estimate

$$\theta_* = \arg\max_\theta \{\log p(\theta|\mathcal{D})\} = \arg\min_\theta \underbrace{\left\{ -\sum_{n=1}^N \log p(\mathbf{y}_n \mid \mathbf{x}_n, \theta) - \log p(\theta) \right\}}_{\mathcal{L}(\theta)}. \tag{1}$$

A Taylor expansion around $\theta_*$ of the regularized loss $\mathcal{L}(\theta)$ then yields

$$\hat{\mathcal{L}}(\theta) \approx \mathcal{L}(\theta_*) + \langle \nabla_\theta \mathcal{L}(\theta)|_{\theta=\theta_*}, (\theta - \theta_*)\rangle + \frac{1}{2}\langle(\theta - \theta_*), \mathrm{H}_\theta[\mathcal{L}](\theta)|_{\theta=\theta_*}(\theta - \theta_*)\rangle, \tag{2}$$

where we know that $\nabla_\theta \mathcal{L}(\theta)|_{\theta=\theta_*} \approx 0$, and $\mathrm{H}_\theta[\mathcal{L}](\theta) \in \mathbb{R}^{K \times K}$ denotes the Hessian of the loss. This expansion suggests that the approximate posterior covariance should be the inverse Hessian $\Sigma = \mathrm{H}_\theta[\mathcal{L}](\theta)|_{\theta=\theta_*}^{-1}$. The marginal likelihood of the data is then approximated as $p(\mathcal{D}) \approx \exp(-\mathcal{L}(\theta_*))(2\pi)^{D/2} \det(\Sigma)^{1/2}$. This is commonly used for training hyper-parameters of both the likelihood and the prior [Immer et al., 2021a, Antorán et al., 2022]. We refer to appendix A for further details.

**Tricks of the trade.** Despite the simplicity of the Laplace approximation, its application to modern neural networks is not trivial. The first issue is that the Hessian matrix is too large to be stored in memory, which is commonly handled by approximately reducing the Hessian to being diagonal, low-rank, Kronecker factored, or only considered for a subset of parameters (see Daxberger et al. [2021a] for a review). Secondly, the Hessian is generally not positive definite [Sagun et al., 2016], which is commonly handled by approximating the Hessian with the generalized Gauss-Newton approximation [Foresee and Hagan, 1997, Schraudolph, 2002]. Furthermore, estimating the predictive distribution using Monte Carlo samples from the Laplace approximated posterior usually performs poorly [Lawrence, 2001, Chapter 5][Ritter et al., 2018] even for small models. Indeed, the Laplace approximation can place probability mass in low regions of the posterior. A solution, already proposed by [Mackay, 1992, Chapter 4], is to consider a first-order Taylor expansion around $\theta_*$, and use the sample to use the "linearized" function $f_\theta^{\text{lin}}(\mathbf{x}) = f_{\theta_*}(\mathbf{x}) + \langle \nabla_\theta f_\theta(\mathbf{x})|_{\theta=\theta_*}, \theta - \theta_*\rangle$ as predictive, where $\nabla_\theta f_\theta(\mathbf{x})|_{\theta=\theta_*} \in \mathbb{R}^{C \times K}$ is the Jacobian. Recently, this approach has been justified by Khan et al. [2019], Immer et al. [2021b], who proved that the generalized Gauss-Newton approximation is the exact Hessian of this new linearized model. Even if this is a linear function with respect to the parameters $\theta$, empirically it achieves better performance than the classic Laplace approximation.

Although not theoretically justified, optimizing the prior precision post-hoc has been shown to play a crucial role in the Laplace approximation [Ritter et al., 2018, Kristiadi et al., 2020, Immer et al., 2021a, Daxberger et al., 2021a]. This is usually done either using cross-validation or by maximizing the log-marginal likelihood. In principle, this regularizes the Hessian, and the associated approximate posterior concentrates around the MAP estimate.

**Strengths & weaknesses.**   The main strength of the Laplace approximation is its simplicity in implementation due to the popularization of automatic differentiation. The Gaussian approximate posterior is, however, quite crude and often does not capture the shape locally of the true posterior [Sagun et al., 2016]. Furthermore, the common reduction of the Hessian to not correlate all model parameters limit the expressive power of the approximate posterior.

# 3   Riemannian Laplace approximations

We aim to construct a parametric approximate posterior that better reflects the local shape of the true posterior and captures nonlinear correlations between parameters. The basic idea is to retain the Laplace approximation but change the parameter space $\Theta$ to locally encode the *training loss*. To realize this idea, we will first endow the parameter space with a suitable Riemannian metric (Sec. 3.1) and then construct a Laplace approximation according to this metric (Sec. 3.2).

## 3.1   A loss-aware Riemannian geometry

For a given parameter value $\theta \in \Theta$, we can measure the training loss $\mathcal{L}(\theta)$ of the associated neural network. Assuming that the loss changes smoothly with $\theta$, we can interpret the loss surface $\mathcal{M} = g(\theta) = [\theta, \mathcal{L}(\theta)] \in \mathbb{R}^{K+1}$ as a $K$-dimensional manifold in $\mathbb{R}^{K+1}$. The goal of Riemannian geometry [Lee, 2019, do Carmo, 1992] is to do calculations that are restricted to such manifolds.

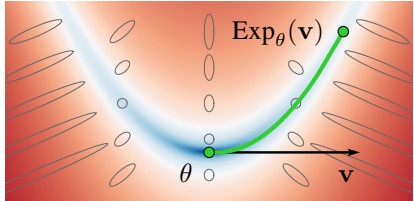

Figure 2: The parameter space $\Theta$ of the BNN together with examples of the Riemannian metric and the exponential map. Note that the Riemannian metric adapts to the shape of the loss which causes the geodesic to follow its shape.

**The metric.**   We can think of the parameter space $\Theta$ as being the *intrinsic coordinates* of the manifold $\mathcal{M}$, and it is beneficial to do all calculations directly in these coordinates. Note that a vector tangential to the manifold can be written as $\mathbf{J}_g(\theta)\mathbf{v} \in \mathbb{R}^{K+1}$, where $\mathbf{J}_g : \Theta \to \mathbb{R}^{K+1 \times K}$ is the Jacobian of $g$ that spans the tangent space $\mathcal{T}_{g(\theta)}\mathcal{M}$ at the point $g(\theta) \in \mathcal{M}$ and $\mathbf{v} \in \mathbb{R}^K$ is the vector of *tangential coordinates* for this basis of the tangent space. We can take inner products between two tangent vectors in the same tangent space as $\langle \mathbf{J}_g(\theta)\mathbf{v}_1, \mathbf{J}_g(\theta)\mathbf{v}_2 \rangle = \mathbf{v}_1^\mathsf{T}\mathbf{J}_g(\theta)^\mathsf{T}\mathbf{J}_g(\theta)\mathbf{v}_2$, which, we note, is now expressed directly in the intrinsic coordinates. From this observation, we define the *Riemannian metric* $\mathbf{M}(\theta) = \mathbf{J}_g(\theta)^\mathsf{T}\mathbf{J}_g(\theta)$, which gives us a notion of a local inner product in the intrinsic coordinates of the manifold (see ellipsoids in Fig. 2). The Jacobian of $g$ is particularly simple $\mathbf{J}_g(\theta) = [\mathbb{I}_K, \nabla_\theta \mathcal{L}]^\mathsf{T}$, such that the metric takes the form

$$\mathbf{M}(\theta) = \mathbb{I}_K + \nabla_\theta \mathcal{L}(\theta)\nabla_\theta \mathcal{L}(\theta)^\mathsf{T}. \tag{3}$$

**The exponential map.**   A local inner product allows us to define the length of a curve $c : [0,1] \to \Theta$ as $\texttt{length}[c] = \int_0^1 \sqrt{\langle \dot{c}(t), \mathbf{M}(c(t))\dot{c}(t)\rangle}\mathrm{d}t$, where $\dot{c}(t) = \partial_t c(t)$ is the velocity. From this, the *distance* between two points can be defined as the length of the shortest connecting curve, where the latter is known as the *geodesic curve*. Such geodesics can be expressed as solutions to a system of second-order non-linear ordinary differential equations (ODEs), which is given in appendix B alongside further details on geometry. Of particular interest to us is the *exponential map*, which solves these ODEs subject to an initial position and velocity. This traces out a geodesic curve with a given starting point and direction (see Fig. 2). Geometrically, we can also think of this as mapping a tangent vector *back to the manifold*, and we write the map as $\mathrm{Exp} : \mathcal{M} \times \mathcal{T}_\theta \mathcal{M} \to \mathcal{M}$.

The tangential coordinates $\mathbf{v}$ can be seen as a coordinate system for the neighborhood around $\theta$, and since the exponential map is locally a bijection we can represent any point locally with a unique tangent vector. However, these coordinates correspond to the tangent space that is spanned by $\mathbf{J}_g(\theta)$, which implies that by changing this basis the associated coordinates change as well. By

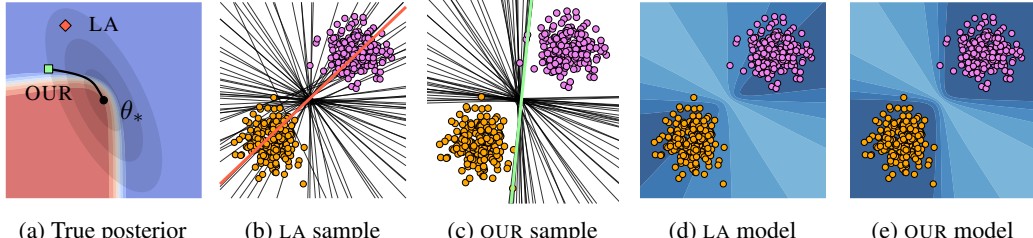

| (a) True posterior | (b) LA sample | (c) OUR sample | (d) LA model | (e) OUR model |

Figure 3: The LA assigns probability mass to regions where the true posterior is nearly zero, and a sample from this region corresponds to a poor classifier. Considering this sample as the initial velocity for the exponential map, the generated sample falls within the true posterior and the associated classifier performs well. As a result, our model quantifies better the uncertainty.

orthonormalizing this basis we get the *normal coordinates* where the metric vanishes. Let $\mathbf{v}$ the tangential coordinates and $\bar{\mathbf{v}}$ the corresponding normal coordinates, then it holds that

$$\langle \mathbf{v}, \mathbf{M}(\theta)\mathbf{v} \rangle = \langle \bar{\mathbf{v}}, \bar{\mathbf{v}} \rangle \Rightarrow \mathbf{v} = \mathbf{A}(\theta)\bar{\mathbf{v}} \quad \text{with} \quad \mathbf{A}(\theta) = \mathbf{M}(\theta)^{-1/2}. \tag{4}$$

We will use the normal coordinates when doing Taylor expansions of the log-posterior, akin to standard Laplace approximations.

## 3.2 The proposed approximate posterior

In order to Taylor-expand the loss according to the metric, we first express the loss in normal coordinates of the tangent space at $\theta_*$, $h(\bar{\mathbf{v}}) = \mathcal{L}(\mathrm{Exp}_{\theta_*}(\mathbf{M}(\theta_*)^{-1/2}\bar{\mathbf{v}}))$. Following the standard Laplace approximation, we perform a second-order Taylor expansion of $h$ as

$$\hat{h}(\bar{\mathbf{v}}) \approx h(0) + \langle \partial_{\bar{\mathbf{v}}} h(\bar{\mathbf{v}})\big|_{\bar{\mathbf{v}}=0}, \bar{\mathbf{v}} \rangle + \frac{1}{2}\langle \bar{\mathbf{v}}, \mathrm{H}_{\bar{\mathbf{v}}}[h](\bar{\mathbf{v}})\big|_{\bar{\mathbf{v}}=0}\bar{\mathbf{v}} \rangle, \tag{5}$$

where $\partial_{\bar{\mathbf{v}}} h(\bar{\mathbf{v}})\big|_{\bar{\mathbf{v}}=0} = \mathbf{A}(\theta_*)^{\mathsf{T}}\nabla_\theta\mathcal{L}(\theta)\big|_{\theta=\theta_*} \approx 0$ as $\theta_*$ minimize the loss and $\mathrm{H}_{\bar{\mathbf{v}}}[h](\bar{\mathbf{v}})\big|_{\bar{\mathbf{v}}=0} = \mathbf{A}(\theta_*)^{\mathsf{T}}\mathrm{H}_\theta[\mathcal{L}](\theta)\mathbf{A}(\theta_*)\big|_{\theta=\theta_*}$ with $\mathrm{H}_\theta[\mathcal{L}](\theta)$ the standard Euclidean Hessian matrix of the loss. Further details about this step can be found in appendix B.

**Tangential Laplace.** Similar to the standard Laplace approximation, we get a Gaussian approximate posterior $\bar{q}(\bar{\mathbf{v}}) = \mathcal{N}(\bar{\mathbf{v}} \mid 0, \overline{\Sigma})$ on the tangent space in the normal coordinates with covariance $\overline{\Sigma} = \mathrm{H}_{\bar{\mathbf{v}}}[h](\bar{\mathbf{v}})\big|_{\bar{\mathbf{v}}=0}^{-1}$. Note that changing the normal coordinates $\bar{\mathbf{v}}$ to tangential coordinates $\mathbf{v}$ is a linear transformation and hence $\mathbf{v} \sim \mathcal{N}(0, \mathbf{A}(\theta_*)\overline{\Sigma}\mathbf{A}(\theta_*)^{\mathsf{T}})$, which means that this covariance is equal to $\mathrm{H}_\theta[\mathcal{L}](\theta)\big|_{\theta=\theta_*}^{-1}$ since $\mathbf{A}(\theta_*)$ is a symmetric matrix, and hence, it cancels out. The approximate posterior $q_{\mathcal{T}}(\mathbf{v}) = \mathcal{N}(\mathbf{v} \mid 0, \Sigma)$ in tangential coordinates, thus, matches the covariance of the standard Laplace approximation.

**The predictive posterior.** We can approximate the predictive posterior distribution using Monte Carlo integration as $p(y|\mathbf{x}', \mathcal{D}) = \int p(y|\mathbf{x}', \mathcal{D}, \theta)q(\theta)\mathrm{d}\theta = \int p(y|\mathbf{x}', \mathcal{D}, \mathrm{Exp}_{\theta_*}(\mathbf{v}))q_{\mathcal{T}}(\mathbf{v})\mathrm{d}\mathbf{v} \approx \frac{1}{S}\sum_{s=1}^{S} p(y|\mathbf{x}', \mathcal{D}, \mathrm{Exp}_{\theta_*}(\mathbf{v}_s))$, $\mathbf{v}_s \sim q_{\mathcal{T}}(\mathbf{v})$. Intuitively, this generates tangent vectors according to the standard Laplace approximation and maps them back to the manifold by solving the geodesic ODE. This lets the Riemannian approximate posterior take shape from the loss landscape, which is largely ignored by the standard Laplace approximation. We emphasize that this is a general construction that applies to the same Bayesian inference problems as the standard Laplace approximation and is not exclusive to Bayesian neural networks.

The above analysis also applies to the linearized Laplace approximation. In particular, when the $f_\theta^{\mathrm{lin}}(\mathbf{x})$ is considered instead of the $f_\theta(\mathbf{x})$ the loss function in (1) changes to $\mathcal{L}^{\mathrm{lin}}(\theta)$. Consequently, our Riemannian metric is computed under this new loss, and $\nabla_\theta\mathcal{L}^{\mathrm{lin}}(\theta)$ appears in the metric (3).

**Example.** To build intuition, we consider a logistic regressor on a linearly separable dataset (Fig. 3). The likelihood of a point $\mathbf{x} \in \mathbb{R}^2$ to be in one class is $p(C = 1|\mathbf{x}) = \sigma(\mathbf{x}^{\mathsf{T}}\theta + b)$, where $\sigma(\cdot)$ is the `sigmoid` function, $\theta \in \mathbb{R}^2$ and $b \in \mathbb{R}$. After learning the parameters, we fix $b_*$ and show the posterior with respect to $\theta$ together with the corresponding standard Laplace approximation (Fig. 3a).

We see that the approximation assigns significant probability mass to regions where the true posterior is near-zero, and the result of a corresponding sample is a poor classifier (Fig. 3b). Instead, when we consider this sample as the initial velocity and compute the associated geodesic with the exponential map, we generate a sample at the tails of the true posterior which corresponds to a well-behaved model (Fig. 3c). We also show the predictive distribution for both approaches and even if both solve easily the classification problem, our model better quantifies uncertainty (Fig. 3e).

### 3.3 Efficient implementation

Our approach is a natural extension of the standard Laplace approximation, which locally adapts the approximate posterior to the true posterior. The caveat is that computational cost increases since we need to integrate an ODE for every sample. We now discuss partial alleviations.

**Integrating the ODE.** In general, the system of second-order nonlinear ODEs (see appendix B for the general form) is non-trivial as it depends on the geometry of the loss surface, which is complicated in the over-parametrized regime [Li et al., 2018]. In addition, the dimensionality of the parameter space is high, which makes the solution of the system even harder. Nevertheless, due to the structure of our Riemannian metric (3), the ODE simplifies to

$$\ddot{c}(t) = -\nabla_\theta \mathcal{L}(c(t)) \left(1 + \nabla_\theta \mathcal{L}(c(t))^\mathsf{T} \nabla_\theta \mathcal{L}(c(t))\right)^{-1} \langle \dot{c}(t), \mathrm{H}_\theta[\mathcal{L}](c(t))\dot{c}(t)\rangle, \qquad (6)$$

which can be integrated reasonably efficiently with standard solvers. In certain cases, this ODE can be further simplified, for example when we consider the linearized loss $\mathcal{L}^{\mathrm{lin}}(\theta)$ and Gaussian likelihood.

**Automatic-differentiation.** The ODE (6) requires computing both gradient and Hessian, which are high-dimensional objects for modern neural networks. While we need to compute the gradient explicitly, we do not need to compute and store the Hessian matrix, which is infeasible for large networks. Instead, we rely on modern automatic-differentiation frameworks to compute the Hessian-vector product between $\mathrm{H}_\theta[\mathcal{L}](c(t))$ and $\dot{c}(t)$ directly. This both reduces memory use, increases speed, and simplifies the implementation.

**Mini-batching.** The cost of computing the metric, and hence the ODE scales linearly with the number of training data, which can be expensive for large datasets. A reasonable approximation is to mini-batch the estimation of the metric when generating samples, i.e. construct a batch $\mathcal{B}$ of $B$ random data points and use the associated loss in the ODE (6). As usual, we assume that $\mathcal{L}(\theta) \approx (N/B)\mathcal{L}_\mathcal{B}(\theta)$. Note that we only mini-batch the metric and not the covariance of our approximate posterior $q_\mathcal{T}(\mathbf{v})$.

We analyze the influence of mini-batching in our methods and provide empirical evidence in Fig. 4. In principle, the geometry of the loss surface $\mathcal{L}(\theta)$ controls the geodesics via the associated Riemannian metric, so when we consider the full dataset we expect the samples to behave similarly to $f_{\theta_*}(\mathbf{x})$. In other words, our approximate posterior generates weights near $\theta_*$ resulting in models with similar or even better loss. When we consider a batch the geometry of the associated loss surface $\mathcal{L}_\mathcal{B}(\theta)$ controls the generated geodesic. So if the batch represents well the structure of the full dataset, then the resulting model will be meaningful with respect to the original problem, and in addition, it may exhibit some variation that is beneficial from the Bayesian perspective for the quantification of the uncertainty. The same concept applies in the linearized version, with the difference that when the full dataset is considered the geometry of $\mathcal{L}^{\mathrm{lin}}(\theta)$ may over-regularize the geodesics. Due to the linear nature of $f_\theta^{\mathrm{lin}}(\theta)$ the associated Riemannian metric is small only close to $\theta_*$ so the generated samples are similar to $f_{\theta_*}(\mathbf{x})$. We relax this behavior and potentially introduce variations in the resulting models when we consider a different batch whenever we generate a sample. Find more details in appendix D.

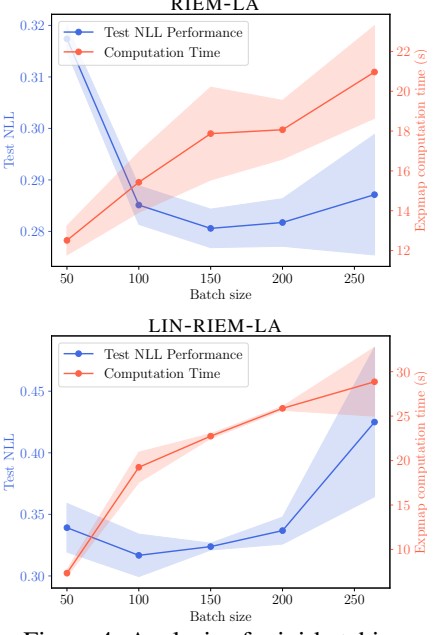

Figure 4: Analysis of mini-batching

## 4    Related work

**Bayesian neural networks.**    Exact inference for BNNs is generally infeasible when the number of parameters is large. Several methods rely on approximate inference, which differs in their trade-off between computational cost and the goodness of the approximation. These techniques are usually based on the Laplace approximation [MacKay, 1992], variational inference [Graves, 2011, Blundell et al., 2015, Khan et al., 2018], dropout [Gal and Ghahramani, 2016], stochastic weight averaging [Izmailov et al., 2018, Maddox et al., 2019] or Monte Carlo based methods [Neal, 1995], where the latter is often more expensive.

**Laplace approximations.**    In this work, we are primarily focused on Laplace approximations, although the general geometric idea can be used in combination with any other inference approach listed above. Particularly, Laplace's method for BNNs was first proposed by Mackay [1992] in his *evidence* framework, where a closed-form approximation of predictive probabilities was also derived. This one uses a first-order Taylor expansion, also known as *linearization* around the MAP estimate. For long, Laplace's method was infeasible for modern architectures with large networks due to the exact computation of the Hessian. The seminal works of Martens and Grosse [2015] and Botev et al. [2017] made it possible to approximate the Hessian of large networks, which made Laplace approximations feasible once more [Ritter et al., 2018]. More recently, the Laplace approximation has become a go-to tool for turning trained neural networks into BNNs in a *post-hoc* manner, thanks to easy-to-use software [Daxberger et al., 2021a] and new approaches to scale up computation [Antorán et al., 2022]. In this direction, other works have only considered a subset of the network parameters [Daxberger et al., 2021b, Sharma et al., 2023], especially the last-layer. This is *de facto* the only current method competitive with *ensembles* [Lakshminarayanan et al., 2017].

**Posterior refinement.**    Much work has gone into building more expressive approximate posteriors. Recently, Kristiadi et al. [2022] proposed to use normalizing flows to get a non-Gaussian approximate distribution using the Laplace approximation as a base distribution. Although this requires training an additional model, they showed that few bijective transformations are enough to improve the last-layer posterior approximation. Immer et al. [2021b], instead, propose to refine the Laplace approximation by using Gaussian variational Bayes or a Gaussian process. This still results in a Gaussian distribution, but it has proven beneficial for linearized Laplace approximations. Other approaches rely on a mixture of distributions to improve the goodness of the approximation. Miller et al. [2017] expand a variational approximation iteratively adding components to a mixture, while Eschenhagen et al. [2021] use a weighted sum of posthoc Laplace approximations generated from different pre-trained networks. Havasi et al. [2021], instead, introduces auxiliary variables to make a local refinement of a mean-field variational approximation.

**Differential geometry.**    Differential geometry is increasingly playing a role in inference. Arvanitidis et al. [2016] make a Riemannian normal distribution locally adapt to data by learning a suitable Riemannian metric from data. In contrast, our metric is derived from the model. This is similar in spirit to work that investigates pull-back metrics in latent variable models [Tosi et al., 2014, Arvanitidis et al., 2018, Hauberg, 2018b]. A similar Riemannian metric has been used as a surrogate for the Fisher metric for Riemannian Hamiltomian Monte-Carlo sampling [Hartmann et al., 2022]. In addition to that, the geometry of the latent parameter space of neural networks was recently analyzed by Kristiadi et al. [2023] focusing on the invariance of flatness measures with respect to re-parametrizations. Finally, we note that Hauberg [2018a] considers Laplace approximations on the sphere as part of constructing a recursive Kalman-like filter.

## 5    Experiments

We evaluate our Riemannian LA (RIEM-LA) using illustrative examples, image datasets where we use a convolutional architecture, and real-world classification problems. We compare our method and its linearized version to standard and linearized LA. All predictive distributions are approximated using Monte Carlo (MC) samples. Although last-layer LA is widely used lately, we focus on approximating the posterior of all the weights of the network. In all experiments, we maximize the marginal log-likelihood to tune the hyperparameters of the prior and the likelihood as proposed in [Daxberger et al., 2021a]. To evaluate the performance in terms of uncertainty estimation we considered the standard metrics in the literature: negative log-likelihood (NLL), the Brier score (BRIER), the expected calibration error (ECE), and the maximum calibration error (MCE). More experiments are

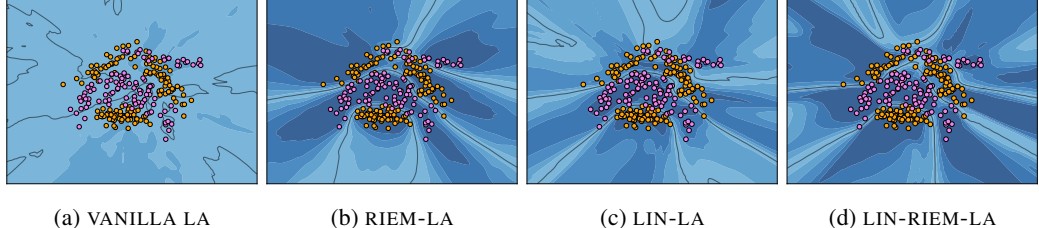

| (a) VANILLA LA | (b) RIEM-LA | (c) LIN-LA | (d) LIN-RIEM-LA |

Figure 6: Binary classification confidence estimate using 100 Monte-Carlo samples on the banana dataset. Vanilla LA underfit, while the other three methods are able to be certain within the data and uncertain far away. Note, for linearized RIEM-LA we solve the expmap using a different subset of the data. Confidence plots of all different methods can be found in the supplementary material. Black lines are the decision boundaries.

available in appendix D together with the complete training and modeling details. In appendix C.2 we analyze the runtime to compute the exponential map over different dataset and model sizes. Code to reproduce the results is publicly available at https://github.com/federicobergamin/riemannian-laplace-approximation

## 5.1 Regression problem

We consider the toy-regression problem proposed by Snelson and Ghahramani [2005]. The dataset contains 200 data points, and we randomly pick 150 examples as our training set and the remaining 50 as a test set. As shown by Lawrence [2001], Ritter et al. [2018], using samples from the LA posterior performs poorly in regression even if the Hessian is not particularly ill-conditioned, i.e. when the prior precision is optimized. For this reason, the linearization approach is necessary for regression with standard LA. Instead, we show that even our basic approach fixes this problem when the prior is optimized. We tested our approach by considering two fully connected networks, one with one hidden layer with 15 units and one with two layers with 10 units each, both with `tanh` activations. Our approach approximates well the true posterior locally, so the resulting function samples follow the data. Of course, if the Hessian is extremely degenerate our approach also suffers, as the initial velocities are huge. When we consider the linearized version of our approach the result is the same as the standard LA-linearization, which we include in the appendix D, where we also report results for in-between uncertainty as proposed by Foong et al. [2019] and a comparison with Hamiltonian Monte Carlo.

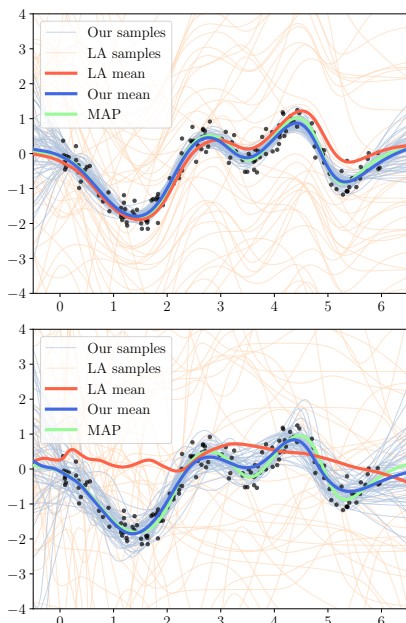

Figure 5: Posterior samples under a simple *(top)* and an overparametrized model *(bottom)*. Vanilla LA is known to generate bad models, while our samples from RIEM-LA quantify well the uncertainty.

## 5.2 Classification problems

**Illustrative example.** We consider a 2-dimensional binary classification problem using the banana dataset which is shown in Fig. 6. We train a 2-layer fully connected neural net with 16 hidden units per layer and `tanh` activation. For all methods, we use 100 MC samples for the predictive distribution.

As in regression, direct samples from the vanilla LA lead to a really poor model (Fig. 6a) with high uncertainty both within and away from the data support. Instead, the other three methods (Fig. 6b-6d) show a better-behaved confidence that decreases outside of the data support. This is also supported by the metrics in Table 1, where remarkably RIEM-LA performs better in terms of NLL and Brier score on a separate test set.

Table 1: In-distribution results in the banana dataset. We use 100 MC samples both for the LA and our variants. ECE and MCE are computed using $M = 10$ bins. We report mean and standard error over 5 different seeds.

| | BANANA DATASET | | | | | |
| | PRIOR OPTIMIZED | | | PRIOR NOT OPTIMIZED | | |
| METHOD | Accuracy ↑ | NLL↓ | Brier↓ | Accuracy ↑ | NLL↓ | Brier↓ |
|---|---|---|---|---|---|---|
| MAP | $86.69 \pm 0.34$ | $0.333 \pm 0.005$ | $0.0930 \pm 0.0015$ | $86.69 \pm 0.34$ | $0.333 \pm 0.005$ | $0.0930 \pm 0.0015$ |
| VANILLA LA | $59.50 \pm 5.07$ | $0.678 \pm 0.009$ | $0.2426 \pm 0.0046$ | $48.85 \pm 2.32$ | $0.700 \pm 0.003$ | $0.2534 \pm 0.0017$ |
| LIN-LA | $86.99 \pm 0.37$ | $0.325 \pm 0.008$ | $0.0956 \pm 0.0023$ | $\mathbf{86.92 \pm 0.40}$ | $0.403 \pm 0.012$ | $0.1196 \pm 0.0044$ |
| RIEM-LA | $\mathbf{87.57 \pm 0.07}$ | $\mathbf{0.287 \pm 0.002}$ | $\mathbf{0.0886 \pm 0.0006}$ | $87.14 \pm 0.20$ | $\mathbf{0.285 \pm 0.001}$ | $\mathbf{0.0878 \pm 0.0006}$ |
| RIEM-LA (BATCHES) | $87.30 \pm 0.08$ | $\mathbf{0.286 \pm 0.001}$ | $0.0890 \pm 0.0000$ | $87.32 \pm 0.17$ | $0.294 \pm 0.002$ | $0.0895 \pm 0.0004$ |
| LIN-RIEM-LA | $87.02 \pm 0.38$ | $0.415 \pm 0.029$ | $0.0967 \pm 0.0024$ | $85.33 \pm 0.31$ | $0.884 \pm 0.037$ | $0.1252 \pm 0.0022$ |
| LIN-RIEM-LA (BATCHES) | $\mathbf{87.77 \pm 0.24}$ | $0.298 \pm 0.006$ | $\mathbf{0.0887 \pm 0.0011}$ | $86.16 \pm 0.21$ | $0.352 \pm 0.002$ | $0.0994 \pm 0.0011$ |

As we discussed in Sec. 3.3, using a subset of the dataset for computing the exponential map can be beneficial for our linearized manifold in addition to speeding up computation. In Fig. 6d we plot the confidence for our linearized approach using batches while in appendix D we show the confidence of the same approach using the full data for solving the ODEs. We can see that our linearized RIEM-LA tends to be overconfident outside the data region and also close to the decision boundary. This behaviour can be found in the high NLL that linearized RIEM-LA gets compared to our vanilla approach and linearized LA.

**UCI datasets.**    We compare our approach against the standard LA on a set of six UCI classification datasets using a fully connected network with a single layer, 50 hidden units and `tanh` activation. The predictive distribution is estimated using MC with 30 samples from the approximate posterior of each approach. In Table 2 we compare the methods in terms of their negative log-likelihood (NLL) in the test set. All other metrics are reported in appendix D. We are considering the setting where we optimize the prior-precision post-hoc, which is the optimal setting for LA and linearized LA. We consider our standard approaches without using batches, which we have seen that specifically for our linearized approach may lead to sub-optimal performance.

From the results in Table 2 we see that our RIEM-LA consistently performs better in terms of negative log-likelihood than vanilla and linearized LA. We also observe that in two datasets the performance of our linearized RIEM-LA is not optimal. This implies that the loss surface of the linearized loss potentially over-regularizes the geodesics as we analyzed in Sec. 3.3, and in this case, considering mini-batching could have been beneficial.

Table 2: Negative log-likelihood (lower is better) on UCI datasets for classification. Predictive distribution is estimated using 30 MC samples. Mean and standard error over 5 different seeds.

| | PRIOR PRECISION OPTIMIZED | | | | |
| DATASET | MAP | VANILLA LA | RIEM-LA | LINEARIZED LA | LINEARIZED RIEM-LA |
|---|---|---|---|---|---|
| VEHICLE | $0.975 \pm 0.081$ | $1.209 \pm 0.020$ | $\mathbf{0.454 \pm 0.024}$ | $0.875 \pm 0.020$ | $\mathbf{0.494 \pm 0.044}$ |
| GLASS | $2.084 \pm 0.323$ | $1.737 \pm 0.037$ | $\mathbf{1.047 \pm 0.224}$ | $1.365 \pm 0.058$ | $1.359 \pm 0.299$ |
| IONOSPHERE | $1.032 \pm 0.175$ | $0.673 \pm 0.013$ | $\mathbf{0.344 \pm 0.068}$ | $0.497 \pm 0.015$ | $0.625 \pm 0.110$ |
| WAVEFORM | $1.076 \pm 0.110$ | $0.888 \pm 0.030$ | $\mathbf{0.459 \pm 0.057}$ | $0.640 \pm 0.002$ | $0.575 \pm 0.065$ |
| AUSTRALIAN | $1.306 \pm 0.146$ | $0.684 \pm 0.011$ | $\mathbf{0.541 \pm 0.053}$ | $\mathbf{0.570 \pm 0.016}$ | $0.833 \pm 0.108$ |
| BREAST CANCER | $\mathbf{0.225 \pm 0.076}$ | $0.594 \pm 0.030$ | $\mathbf{0.176 \pm 0.092}$ | $0.327 \pm 0.022$ | $\mathbf{0.202 \pm 0.073}$ |

**Image classification.**    We consider a small convolutional neural network on MNIST and FashionMNIST. Our network consists of two convolutional layers followed by average pooling layers and three fully connected layers. We consider a model of this size as the high dimensionality of the parameter space is one of the main limitations of the ODE solver. For the training of the model, we subsample each dataset and we consider 5000 observations by keeping the proportionality of labels, and we test in the full test set containing 8000 examples. In Table 3 we compare the different methods with the prior precision optimized as this is the ideal setting for the linearized LA. We refer to appendix D for the setting with the prior precision not optimized.

From the results we observe that our standard RIEM-LA performs better than all the other methods in terms of NLL and Brier score, meaning that the models are better calibrated, but it also leads to a more accurate classifier than the MAP. In terms of ECE, it seems that considering the linearized

Table 3: Image classification results using a CNN on MNIST and FashionMNIST. The network is trained on 5000 examples and we test the in-distribution performance on the test set, which contains 8000 examples. We use 25 Monte Carlo samples to approximate the predictive distribution and 1000 datapoints per batch in our batched manifolds. Calibration metrics are computed using $M = 15$ bins. We report mean and standard error for each metric over 3 different seeds.

| METHOD | CNN ON MNIST - PRIOR PRECISION OPTIMIZED | | | | |
| --- | --- | --- | --- | --- | --- |
| | Accuracy ↑ | NLL↓ | Brier↓ | ECE↓ | MCE↓ |
| MAP | $95.02 \pm 0.17$ | $0.167 \pm 0.005$ | $0.0075 \pm 0.0002$ | $1.05 \pm 0.14$ | $39.94 \pm 14.27$ |
| VANILLA LA | $88.69 \pm 1.84$ | $0.871 \pm 0.026$ | $0.0393 \pm 0.0013$ | $42.11 \pm 1.22$ | $50.52 \pm 1.45$ |
| LIN-LA | $94.91 \pm 0.26$ | $0.204 \pm 0.006$ | $0.0087 \pm 0.0003$ | $6.30 \pm 0.08$ | $39.30 \pm 16.77$ |
| RIEM-LA | $\mathbf{96.74 \pm 0.12}$ | $\mathbf{0.115 \pm 0.003}$ | $\mathbf{0.0052 \pm 0.0002}$ | $2.48 \pm 0.06$ | $38.03 \pm 15.02$ |
| RIEM-LA (BATCHES) | $95.67 \pm 0.19$ | $0.170 \pm 0.005$ | $0.0072 \pm 0.0002$ | $5.40 \pm 0.06$ | $22.40 \pm 0.51$ |
| LIN-RIEM-LA | $95.44 \pm 0.18$ | $0.149 \pm 0.004$ | $0.0068 \pm 0.0003$ | $\mathbf{0.66 \pm 0.03}$ | $39.40 \pm 14.75$ |
| LIN-RIEM-LA (BATCHES) | $95.14 \pm 0.20$ | $0.167 \pm 0.004$ | $0.0076 \pm 0.0002$ | $3.23 \pm 0.04$ | $\mathbf{18.10 \pm 2.50}$ |

| METHOD | CNN ON FASHIONMNIST - PRIOR PRECISION OPTIMIZED | | | | |
| --- | --- | --- | --- | --- | --- |
| | Accuracy ↑ | NLL↓ | Brier↓ | ECE↓ | MCE↓ |
| MAP | $79.88 \pm 0.09$ | $0.541 \pm 0.002$ | $0.0276 \pm 0.0000$ | $\mathbf{1.66 \pm 0.07}$ | $24.07 \pm 1.50$ |
| VANILLA LA | $74.88 \pm 0.83$ | $1.026 \pm 0.046$ | $0.0482 \pm 0.0019$ | $31.63 \pm 1.28$ | $43.61 \pm 2.95$ |
| LIN-LA | $79.85 \pm 0.13$ | $0.549 \pm 0.001$ | $0.0278 \pm 0.0001$ | $3.23 \pm 0.44$ | $37.88 \pm 17.98$ |
| RIEM-LA | $\mathbf{83.33 \pm 0.17}$ | $\mathbf{0.472 \pm 0.001}$ | $\mathbf{0.0237 \pm 0.0001}$ | $3.13 \pm 0.48$ | $10.94 \pm 2.11$ |
| RIEM-LA (BATCHES) | $81.65 \pm 0.18$ | $0.525 \pm 0.004$ | $0.0263 \pm 0.0002$ | $5.80 \pm 0.73$ | $35.30 \pm 18.40$ |
| LIN-RIEM-LA | $81.33 \pm 0.10$ | $0.521 \pm 0.004$ | $0.0261 \pm 0.0002$ | $\mathbf{1.59 \pm 0.40}$ | $25.53 \pm 0.10$ |
| LIN-RIEM-LA (BATCHES) | $80.49 \pm 0.13$ | $0.529 \pm 0.003$ | $0.0269 \pm 0.0002$ | $2.10 \pm 0.42$ | $\mathbf{6.14 \pm 1.42}$ |

approach is beneficial in producing better-calibrated models in both datasets. This holds both for our approach linearized RIEM-LA and the standard LA. Optimizing the prior precision post-hoc is crucial for the vanilla LA and associated results can be seen in appendix D. Instead, both our methods appear to be robust and consistent, as they achieve similar performance no matter if the prior precision is optimized or not.

Note that for the mini-batches for our approaches, we consider 20% of the data by randomly selecting 1000 observations while we respect the label frequency based on the full dataset. Clearly, the batch-size is a hyperparameter for our methods and can be estimated systematically using cross-validation. Even if we do not optimize this hyperparameter, we see that our batched version of RIEM-LA and LIN-RIEM-LA perform better than the standard LA and on-par with our LIN-RIEM-LA without batches, implying that a well-tuned batch-size can potentially further improve the performance. Nevertheless, this also shows that our method is robust with respect to the batch-size.

## 6   Conclusion and future directions

We propose an extension to the standard Laplace approximation, which leverages the natural geometry of the parameter space. Our method is parametric in the sense that a Gaussian distribution is estimated using the standard Laplace approximation, but it adapts to the true posterior through a nonparametric Riemannian metric. This is a general mechanism that, in principle, can also apply to, e.g., variational approximations. In a similar vein, while the focus of our work is on Bayesian neural networks, nothing prevents us from applying our method to other model classes.

Empirically, we find that our Riemannian Laplace approximation is better or on par with alternative Laplace approximations. The standard Laplace approximation crucially relies on both linearization and on a fine-tuned prior to give useful posterior predictions. Interestingly, we find that the Riemannian Laplace approximation requires neither. This could suggest that the standard Laplace approximation has a rather poor posterior fit, which our adaptive approach alleviates.

**Limitations.**   The main downside of our approach is the computational cost involved in integrating the ODE, which is a common problem in computational geometry [Arvanitidis et al., 2019]. The cost of evaluating the ODE scales linearly with the number of observations, and in particular it is $O(SWN)$, where $S$ is the number of steps of the solver, $W$ is the number of model parameters and $N$ is the dataset size. Indeed, the solver needs all the training data points to compute both gradient and hvp at each step. We refer to appendix C.1 for a detailed explanation. For big datasets, we have considered

the "obvious" trick of using a random (small) batch of the data when solving the ODE to reduce the complexity. Empirically, we find that this introduces some stochasticity in the sampling, which sometimes is beneficial to explore the posterior distribution better and eventually boost performance, motivating further research. The computational cost also grows with the dimensionality of the parameter space, as the number of necessary solver steps increases, as well as the cost of the Hessian vector products. Our implementation relies on an off-the-shelf ODE solver which runs on the CPU while our automatic-differentiation based approach runs on the GPU, which is inefficient. We expect significant improvements by using an ODE solver that runs exclusively on GPU, while tailor-made numerical integration methods are also of particular interest.

**Future directions.** We showed that equipping the weight-space with a simple Riemannian metric is a promising approach that solves some of the classic LA issues. We believe that this opens up different interesting research directions that either aim to use different metrics instead of the one used in this work or to make this method scale to bigger dataset. Regarding the metric, a possible extension of this work would be to consider the Fisher information matrix in the parameter space. Potential ideas to improve efficiency is to consider approximations of the Riemannian metric leading to simpler ODE systems, for example by using the KFAC instead of the full-Hessian in (6). Another directions, instead, would be to focus on developing a better solver. Indeed, we know the structure and behavior of our ODE system, i.e. geodesics start from low loss which increases along the curve, therefore, a potential direction would be to develop solvers that exploit this information. Usually, general purpose solvers aim for accuracy, while in our case even inexact solutions could be potentially useful if computed fast.

## Acknowledgments and Disclosure of Funding

This work was funded by the Innovation Fund Denmark (0175-00014B) and the Novo Nordisk Foundation through the Center for Basic Machine Learning Research in Life Science (NNF20OC0062606). It also received funding from the European Research Council (ERC) under the European Union's Horizon 2020 research, innovation programme (757360). SH was supported in part by a research grant (42062) from VILLUM FONDEN.

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

# Riemannian Laplace approximations for Bayesian neural networks (Appendix)

## Contents of the Appendix

## A   Laplace background

We provide a more detailed introduction to Laplace approximation and discuss the optimiziation of the prior and likelihood hyperparameters by maximizing the marginal likelihood. In BNNs, Laplace approximation is use to approximate the posterior distribution, i.e.:

$$p(\theta \mid \mathcal{D}) = \frac{p(\mathcal{D} \mid \theta)p(\theta)}{p(\mathcal{D})} = \frac{p(\mathcal{D} \mid \theta)p(\theta)}{\int_{\Theta} p(\mathcal{D} \mid \theta)p(\theta)} = \frac{1}{Z}p(\mathcal{D} \mid \theta)p(\theta). \tag{A.1}$$

This is done by fitting a Gaussian approximation to the unnormalized distribution $p(\mathcal{D} \mid \theta)p(\theta)$ at its peak, where $p(\mathcal{D} \mid \theta)$ is the likelihood distribution and $p(\theta)$ is the prior over the weights. In standard training of neural networks the mean-squared error loss is usually used for regression. This corresponds to optimizing a Gaussian log-likelihood up to a scaling factor while using the cross-entropy loss in classification correspond to minimizing the negative log-likelihood. The usual weight decay, or L2 regularization, corresponds instead to a Gaussian prior $p(\theta)$ distribution. More specifically, training with $\lambda ||\theta||_2^2$ corresponds to a Gaussian prior $\mathcal{N}(\theta; 0, \frac{\lambda^{-2}}{2}I)$

Therefore a natural peak to choose is given by the $\theta_*$, which is the set of weights obtained at the end of the training. Indeed, $\theta_*$ is usually computed by:

$$\theta_* = \arg\min_{\theta} \underbrace{\left\{ -\sum_{n=1}^{N} \log p(\mathbf{y}_n \mid \mathbf{x}_n, \theta) - \log p(\theta) \right\}}_{\mathcal{L}(\theta)}. \tag{A.2}$$

Once we have $\theta_*$, the Laplace approximation uses a a second-order Taylor expansion of $\mathcal{L}(\theta)$ around $\theta_*$, which yields:

$$\hat{\mathcal{L}}(\theta) \approx \mathcal{L}(\theta_*) + \langle \nabla_\theta \mathcal{L}(\theta)|_{\theta=\theta_*}, (\theta - \theta_*) \rangle + \frac{1}{2} \langle (\theta - \theta_*), \mathrm{H}_\theta[\mathcal{L}](\theta)|_{\theta=\theta_*} (\theta - \theta_*) \rangle, \quad (A.3)$$

where the first-order term $\nabla_\theta \mathcal{L}(\theta)|_{\theta=\theta_*} \approx 0$ because the gradient at $\theta_*$ is 0.

By looking at $\mathcal{L}(\theta)$, we can notice that the Hessian is composed by two terms: a data fitting term and a prior term. Assuming $p(\theta) = \mathcal{N}(\theta; 0, \gamma^2 I)$, then the Hessian can be expressed as

$$\mathrm{H}_\theta[\mathcal{L}](\theta)|_{\theta=\theta_*} = \gamma^{-2} \mathbf{I} + \sum_{i=1}^{n} \nabla_\theta^2 \log p(y_i \mid x_i)|_{\theta_*} = (H + \alpha \mathbf{I}), \quad (A.4)$$

where we defined $\alpha = \frac{1}{\gamma^2}$, i.e. the prior precision.

Using the fact that $\hat{\mathcal{L}}(\theta)$ is the negative log-numerator of (A.1), we can get $p(\mathcal{D} \mid \theta)p(\theta)$ by taking the exponential of $-\hat{\mathcal{L}}(\theta)$. By doing so we have:

$$p(\mathcal{D} \mid \theta)p(\theta) \approx \exp(-\hat{\mathcal{L}}(\theta)) = \exp(-\mathcal{L}(\theta_*) - \frac{1}{2}(\theta - \theta_*)^T (\mathbf{H} + \alpha \mathbf{I})(\theta - \theta_*)). \quad (A.5)$$

For simplicity, we can define $\Sigma = (\mathbf{H} + \alpha \mathbf{I})^{-1}$ and rewrite the equation above to obtain:

$$p(\mathcal{D} \mid \theta)p(\theta) \approx \exp(-\mathcal{L}(\theta_*)) \exp(-\frac{1}{2}(\theta - \theta_*)^T \Sigma^{-1}(\theta - \theta_*))) \quad (A.6)$$

We can then use this approximation to estimate the normalizing constant of our approximate posterior, which corresponds to the marginal log-likelihood $p(\mathcal{D})$:

$$p(\mathcal{D}) = Z \approx \int_\Theta \exp(-\mathcal{L}(\theta_*)) \exp(-\frac{1}{2}(\theta - \theta_*)^T \Sigma^{-1}(\theta - \theta_*))) d\theta. \quad (A.7)$$

This can be rewritten as

$$p(\mathcal{D}) \approx \exp(-\mathcal{L}(\theta_*)) \int_\Theta \exp(-\frac{1}{2}(\theta - \theta_*)^T \Sigma^{-1}(\theta - \theta_*))), \quad (A.8)$$

and by using the Gaussian integral properties we can write it as:

$$p(\mathcal{D}) \approx \exp(-\mathcal{L}(\theta_*))(2\pi)^{d/2}(\det \Sigma)^{1/2}, \quad (A.9)$$

and by taking the logarithm we get

$$\log p(\mathcal{D}) \approx -\mathcal{L}(\theta_*) + \log(2\pi)^{d/2} + \log(\det \Sigma)^{1/2}, \quad (A.10)$$

which is the approximation of the log marginal likelihood that we want to maximize to optimize the parameters that appears in $\mathcal{L}(\theta_*)$. In a regression problem, we are interested in optimizing both the variance of the Gaussian likelihood and the prior precision. In classification, instead, we just have the prior precision as a hyperparameter to tune.

# B  Riemannian geometry

We rely on Riemannian geometry [Lee, 2019, do Carmo, 1992] in order to construct our approximate posterior. In a nutshell, a $d$-dimensional Riemannian manifold can be seen intuitively as a smooth $d$-dimensional surface that lies within a Euclidean space of dimension $D > d$, which allows one to compute distances between points that respect the geometry of the surface.

**Definition B.1.** A Riemannian manifold $\mathcal{M}$ is a smooth manifold together with a Riemannian metric $\mathbf{M}(\mathbf{x})$ that acts on the associated tangent space $\mathcal{T}_\mathbf{x} \mathcal{M}$ at any point $\mathbf{x} \in \mathcal{M}$.

**Definition B.2.** A Riemannian metric $\mathbf{M} : \mathcal{M} \to \mathbb{R}^{\dim(\mathcal{M}) \times \dim(\mathcal{M})}$ is a smoothly changing positive definite metric tensor that defines an inner product on the tangent space $\mathcal{T}_\mathbf{x} \mathcal{M}$ at any point $\mathbf{x} \in \mathcal{M}$.

We focus on the Bayesian neural network framework where $\Theta = \mathbb{R}^K$ is the parameter space of the associated deep network, and we consider the manifold $\mathcal{M} = g(\theta) = [\theta, \mathcal{L}(\theta)]$. This is essentially the loss surface which is $K$-dimensional and lies within a $(K+1)$-dimensional Euclidean space. In order to satisfy the smoothness condition for $\mathcal{M}$ we restrict to activation functions as the `tanh`, while common loss function as the mean squared error and softmax are also smooth.

The parameter space $\Theta$ is a parametrization of this surface and technically represents the *intrinsic coordinates* of the manifold, which is known as the *global chart* in the literature. The Jacobian $\mathbf{J}_g(\theta) \in \mathbb{R}^{K+1 \times K}$ of the map $g$ spans the tangent space on the manifold, and a tangent vector can be written as $\mathbf{J}_g(\theta)\mathbf{v}$, where $\mathbf{v} \in \mathbb{R}^K$ are the *tangential coordinates*. We can thus compute the inner product between two tangent vectors in the ambient space using the Euclidean metric therein as

$$\langle \mathbf{J}_g(\theta)\mathbf{v}, \mathbf{J}_g(\theta)\mathbf{u} \rangle = \langle \mathbf{v}, \mathbf{M}(\theta)\mathbf{u} \rangle. \tag{B.1}$$

Here the matrix $\mathbf{M}(\theta) = \mathbf{J}_g(\theta)^{\intercal}\mathbf{J}_g(\theta) = \mathbb{I}_K + \nabla_\theta \mathcal{L}(\theta)\nabla_\theta \mathcal{L}(\theta)^{\intercal}$ is a Riemannian metric that is known in the literature as the *pull-back metric*. Note that the flat space $\Theta$ is technically a smooth manifold and together with $\mathbf{M}(\theta)$ is transformed into a Riemannian manifold. This can be also seen as the *abstract manifold* definition where we only need the intrinsic coordinates and the Riemannian metric to compute geometric quantities, and not the actual embedded manifold in the ambient space.

One example of a geometric quantity is the shortest path between two points $\theta_1, \theta_2 \in \Theta$. Let a curve $c : [0, 1] \to \Theta$ with $c(0) = \theta_1$ and $c(1) = \theta_2$, the its length defined under the Riemannian metric as $\texttt{length}[c] = \int_0^1 \sqrt{\langle \dot{c}(t), \mathbf{M}(c(t))\dot{c}(t) \rangle} dt$. This quantity is computed intrinsically but it also corresponds to the length of the associated curve on $\mathcal{M}$. The shortest path then is defined as $c^*(t) = \arg\min_c \texttt{length}[c]$, but as the length is invariant under re-parametrizations of time, we consider the energy functional instead, which we optimize using the Euler-Lagrange equations. This gives the following system of second-order nonlinear ordinary differential equations (ODEs)

$$\ddot{c}(t) = -\frac{\mathbf{M}^{-1}(c(t))}{2} \left[ 2 \left[ \frac{\partial \mathbf{M}(c(t))}{\partial c_1(t)}, \dots, \frac{\partial \mathbf{M}(c(t))}{\partial c_K(t)} \right] - \frac{\partial \text{vec}[\mathbf{M}(c(t))]^{\intercal}}{\partial c(t)} \right] (\dot{c}(t) \otimes \dot{c}(t)), \quad \text{(B.2)}$$

where $\text{vec}[\cdot]$ stacks the columns of a matrix and $\otimes$ the Kronecker product [Arvanitidis et al., 2018]. A curve that satisfies this system is known as *geodesic* and is potentially the shortest path. When the system is solved as a Boundary Value Problem (BVP) with initial condition $c(0) = \theta_1$ and $c(1) = \theta_2$, we get the geodesic that connects these two points. Let $\mathbf{v} = \dot{c}(0)$ be the velocity of this curve at $t = 0$. When the system is solved as an Initial Value Problem (IVP) with conditions $c(0) = \theta_1$ and $\dot{c}(0) = \mathbf{v}$, we get the geodesic $c_{\mathbf{v}}(t)$ between $c_{\mathbf{v}}(0) = \theta_1$ and $c_{\mathbf{v}}(1) = \theta_2$. This operation is known as the *exponential map* and we use it for our approximate posterior.

In general, an analytic solution for this system of ODEs does not exist [Hennig and Hauberg, 2014, Arvanitidis et al., 2019], and hence, we rely on an approximate numerical off-the-shelf ODE solver. Note that this is a highly complicated system, especially when the Riemannian metric depends on a finite data set, while it is computationally expensive to evaluate it. As we show below, the structure of the Riemannian metric that we consider, allows us to simplify significantly this system.

**Lemma B.3.** *For the Riemannian metric in* (3) *the general ODEs system in* (B.2) *becomes*

$$\ddot{c}(t) = -\frac{\nabla_\theta \mathcal{L}(c(t))}{1 + \langle \nabla_\theta \mathcal{L}(c(t)), \nabla_\theta \mathcal{L}(c(t)) \rangle} \langle \dot{c}(t), \mathbf{H}_\theta[\mathcal{L}](c(t))\dot{c}(t) \rangle \tag{B.3}$$

*Proof.* We consider the general system, and we compute each term individually. To simplify notation we will use $\mathbf{M} := \mathbf{M}(c(t))$, $\nabla := \nabla_\theta \mathcal{L}(c(t))$, $\mathbf{H} := \mathbf{H}_\theta[\mathcal{L}](c(t))$, $\mathbf{H}_i$ the $i$-th column of the Hessian and $\nabla_i$ the $i$-th element of the gradient $\nabla$.

Using the Sherman–Morrison formula we have that

$$\mathbf{M}(c(t))^{-1} = \mathbb{I}_K - \frac{\nabla_\theta \mathcal{L}(c(t))\nabla_\theta \mathcal{L}(c(t))^{\intercal}}{1 + \langle \nabla_\theta \mathcal{L}(c(t)), \nabla_\theta \mathcal{L}(c(t)) \rangle} \tag{B.4}$$

The first term in the brackets is

$$2 \left[ \frac{\partial \mathbf{M}(c(t))}{\partial c_1(t)}, \dots, \frac{\partial \mathbf{M}(c(t))}{\partial c_K(t)} \right] = 2 \left[ \mathbf{H}_1 \nabla^{\intercal} + \nabla \mathbf{H}_1^{\intercal}, \dots, \mathbf{H}_K \nabla^{\intercal} + \nabla \mathbf{H}_K^{\intercal} \right]_{D \times D^2} \tag{B.5}$$

and the second term in the brackets is

$$\frac{\partial \mathrm{vec}[\mathbf{M}(c(t))]^\mathsf{T}}{\partial c(t)} = [\nabla_1 \mathrm{H} + \mathrm{H}_1 \nabla^\mathsf{T}, \dots, \nabla_K \mathrm{H} + \mathrm{H}_K \nabla^\mathsf{T}] \tag{B.6}$$

and their difference is equal to

$$[2\nabla \mathrm{H}_1^\mathsf{T} + \mathrm{H}_1 \nabla^\mathsf{T} - \nabla_1 \mathrm{H}, \dots, 2\nabla \mathrm{H}_K^\mathsf{T} + \mathrm{H}_K \nabla^\mathsf{T} - \nabla_K \mathrm{H}]. \tag{B.7}$$

We compute the matrix-vector product between the matrix (B.7) and the Kronocker product $\dot{c}(t) \otimes \dot{c}(t) = [\dot{c}_1 \dot{c}, \dots, \dot{c}_K \dot{c}]^\mathsf{T} \in \mathbb{R}^{D^2 \times 1}$ which gives

$$\sum_{i=1}^K 2\nabla \mathrm{H}_i^\mathsf{T} \dot{c} \dot{c}_i + \mathrm{H}_i \nabla^\mathsf{T} \dot{c} \dot{c}_i - \nabla_i \mathrm{H} \dot{c} \dot{c}_i = 2\nabla \sum_{i=1}^K \dot{c}_i \mathrm{H}_i^\mathsf{T} \dot{c} + \nabla^\mathsf{T} \dot{c} \sum_{i=1}^K \mathrm{H}_i \dot{c}_i - \mathrm{H} \dot{c} \sum_{i=1}^K \nabla_i \dot{c}_i = 2\nabla\langle \dot{c}, \mathrm{H}\dot{c}\rangle \tag{B.8}$$

where we used that $\sum_{i=1}^K \mathrm{H}_i \dot{c}_i = \mathrm{H}\dot{c}$ and $\sum_{i=1}^K \nabla_i \dot{c}_i = \nabla^\mathsf{T} \dot{c}$. As a final step, we plug-in this result and the inverse of the metric in the general system which gives

$$\ddot{c} = -\frac{1}{2}\left(\mathbb{I}_K - \frac{\nabla \nabla^\mathsf{T}}{1 + \langle \nabla, \nabla\rangle}\right) 2\nabla\langle \dot{c}, \mathrm{H}\dot{c}\rangle = -\frac{\nabla}{1 + \langle \nabla, \nabla\rangle}\langle \dot{c}, \mathrm{H}\dot{c}\rangle. \tag{B.9}$$

$\square$

**Taylor expansion.** As regards the *Taylor expansion* we consider the space $\Theta$ and the Riemannian metric therein $\mathbf{M}(\theta)$ and an arbitrary smooth function $f : \Theta \to \mathbb{R}$. If we ignore the Riemannian metric the second-order approximation of the function $f$ around a point $\mathbf{x} \in \Theta$ is known to be

$$\hat{f}_{\mathrm{Eucl}}(\mathbf{x} + \mathbf{v}) \approx f(\mathbf{x}) + \langle \nabla_\theta f(\theta)\big|_{\theta=\mathbf{x}}, \mathbf{v}\rangle + \frac{1}{2}\langle \mathbf{v}, \mathrm{H}_\theta[f](\theta)\big|_{\theta=\mathbf{x}} \mathbf{v}\rangle, \tag{B.10}$$

where $\nabla_\theta f(\theta)\big|_{\theta=\mathbf{x}}$ is the vector with the partial derivatives evaluated at $\mathbf{x}$ and $\mathrm{H}_\theta[f](\theta)\big|_{\theta=\mathbf{x}}$ the corresponding Hessian matrix with the partial derivatives $\frac{\partial^2 f(\theta)}{\partial \theta_i \partial \theta_j}\big|_{\theta=\mathbf{x}}$. When we take into account the Riemannian metric, then the approximation becomes

$$\hat{f}_{\mathrm{Riem}}(\mathbf{x} + \mathbf{v}) \approx f(\mathbf{x}) + \langle \nabla_\theta f(\theta)\big|_{\theta=\mathbf{x}}, \mathbf{v}\rangle + \frac{1}{2}\langle \mathbf{v}, \left[\mathrm{H}_\theta[f](\theta) - \Gamma_{ij}^k \nabla_\theta f(\theta)_k\right]\big|_{\theta=\mathbf{x}} \mathbf{v}\rangle, \tag{B.11}$$

where $\Gamma_{ij}^k$ are the Christoffel symbols and the Einstein summation is used. Note that even if the Hessian is different, the approximation again is a quadratic function.

Now we consider the Taylor expansion on the associated tangent space at the point $\mathbf{x}$ instead of directly on the parameter space $\Theta$. We define the function $h(\mathbf{v}) = f(\mathrm{Exp}_\mathbf{x}(\mathbf{v}))$ on the tangent space centered at $\mathbf{x}$ and we get that

$$\hat{h}(\mathbf{u}) \approx h(0) + \langle \partial_\mathbf{v} f(\mathrm{Exp}_\mathbf{x}(\mathbf{v}))\big|_{\mathbf{v}=0}, \mathbf{u}\rangle \tag{B.12}$$

$$+ \frac{1}{2}\langle \mathbf{u}, \left[\mathrm{H}_\mathbf{v}[f](\mathrm{Exp}_\mathbf{x}(\mathbf{v})) - \Gamma_{ij}^k \partial_\mathbf{v} f(\mathrm{Exp}_\mathbf{x}(\mathbf{v}))_k\right]\big|_{\mathbf{v}=0} \mathbf{u}\rangle, \tag{B.13}$$

where we apply the chain-rule and we use the fact that $\partial_\mathbf{v} \mathrm{Exp}_\mathbf{x}(\mathbf{v})\big|_{\mathbf{v}=0} = \mathbb{I}$ and $\partial_\mathbf{v}^2 \mathrm{Exp}_\mathbf{x}(\mathbf{v})\big|_{\mathbf{v}=0} = 0$. So, we get that $\partial_\mathbf{v} f(\mathrm{Exp}_\mathbf{x}(\mathbf{v}))\big|_{\mathbf{v}=0} = \nabla_\theta f(\theta)\big|_{\theta=\mathbf{x}}$ and $\mathrm{H}_\mathbf{v}[f](\mathrm{Exp}_\mathbf{x}(\mathbf{v}))\big|_{\mathbf{v}=0} = \frac{\partial^2 f(\theta)}{\partial \theta_i \partial \theta_j}\big|_{\theta=\mathbf{x}}$. The difference here is that the quadratic function is defined on the tangent space, and the exponential map is a non-linear mapping. Therefore, the actual approximation on the parameter space $\hat{f}_{\mathrm{Tangent}}(\mathrm{Exp}_\mathbf{x}(\mathbf{v})) = \hat{h}(\mathbf{v})$ is not a quadratic function as before, but it adapts to the structure of the Riemannian metric. Intuitively, the closer a point is to the base point $\mathbf{x}$ with respect to the Riemannian distance, the more similar is the associated value $\hat{f}_{\mathrm{Tangent}}(\mathrm{Exp}_\mathbf{x}(\mathbf{v}))$ to $f(\mathbf{x})$. In our problem of interest, this behavior is desirable implying that if a parameter $\theta'$ is connected through low-loss regions with a continuous curve to $\theta_*$, then we will assign to $\theta$ high approximate posterior density.

We can easily consider the approximation on the normal coordinates, where we know that the Christoffel symbols vanish, by using the relationship $\mathbf{u} = \mathbf{A}\bar{\mathbf{u}}$ with $\mathbf{A} = \mathbf{M}(\mathbf{x})^{-1/2}$, and thus the Taylor approximation of the function $\bar{h}(\bar{\mathbf{u}}) = f(\mathrm{Exp}_{\mathbf{x}}(\mathbf{A}\bar{\mathbf{u}}))$ becomes

$$\hat{\bar{h}}(\bar{\mathbf{u}}) \approx \hat{h}(0) + \langle \mathbf{A}^{\mathsf{T}} \nabla_\theta f(\theta)\big|_{\theta=\mathbf{x}}, \mathbf{A}\bar{\mathbf{u}}\rangle + \frac{1}{2}\langle \bar{\mathbf{u}}, \mathbf{A}^{\mathsf{T}} \mathrm{H}_\theta[f](\theta)\big|_{\theta=\mathbf{x}}\mathbf{A}\bar{\mathbf{u}}\rangle. \tag{B.14}$$

Further details can be found in related textbooks [Absil et al., 2008] and articles [Pennec, 2006].

**Linearized manifold.** Let us consider a regression problem with likelihood $p(\mathbf{y}|\mathbf{x},\theta) = \mathcal{N}(\mathbf{y}|f_\theta(\mathbf{x}),\sigma^2)$, where $f_\theta$ is a deep neural network, and prior $p(\theta) = \mathcal{N}(\theta|0,\lambda\mathbb{I}_K)$. The loss of the $f_\theta^{\mathrm{lin}}(\mathbf{x})$ is then defined as

$$\mathcal{L}^{\mathrm{lin}}(\theta) = \frac{1}{2\sigma^2}\sum_{n=1}^N (\mathbf{y} - f_{\theta_*}(\mathbf{x}_n) - \langle \nabla_\theta f(\mathbf{x}_n)\big|_{\theta=\theta_*}, \theta - \theta_*\rangle)^2 + \lambda||\theta||^2. \tag{B.15}$$

The gradient and the Hessian of this loss function can be easily computed as

$$\nabla_\theta \mathcal{L}^{\mathrm{lin}}(\theta) = \frac{1}{\sigma^2}\sum_{n=1}^N -(\mathbf{y} - f_{\theta_*}(\mathbf{x}_n) - \langle \nabla_\theta f(\mathbf{x}_n)\big|_{\theta=\theta_*}, \theta - \theta_*\rangle)\nabla_\theta f(\mathbf{x}_n)\big|_{\theta=\theta_*} + 2\lambda\theta \quad \text{(B.16)}$$

$$\mathrm{H}_\theta[\mathcal{L}^{\mathrm{lin}}](\theta) = \frac{1}{\sigma^2}\sum_{n=1}^N \nabla_\theta f(\mathbf{x}_n)\big|_{\theta=\theta_*}\nabla_\theta f(\mathbf{x}_n)\big|_{\theta=\theta_*}^{\mathsf{T}} + 2\lambda\mathbb{I}_K, \tag{B.17}$$

which can be used to evaluate the ODEs system in (6). A similar result can be derived for the binary cross entropy loss and the Bernoulli likelihood.

# C   Implementation details

In this section we present the implementation details of our work. The code will be released upon acceptance.

**Gradient, Hessian, and Jacobian computations.** The initial velocities used for our methods are samples from the Laplace approximation. We rely in the `Laplace` library [Daxberger et al., 2021a] for fitting the Laplace approximation and for optimizing the hyperparameters by using the marginal log-likelihood. We also used the same library to implement all the baselines consider in this work. As we have seen from Sec. 3.3, to integrate the ODE we have to compute (6), which we report also here for clarity:

$$\ddot{c}(t) = -\nabla_\theta \mathcal{L}(c(t))\left(1 + \nabla_\theta \mathcal{L}(c(t))^{\mathsf{T}} \nabla_\theta \mathcal{L}(c(t))\right)^{-1}\langle \dot{c}(t), \mathrm{H}_\theta[\mathcal{L}](c(t))\dot{c}(t)\rangle. \tag{C.1}$$

We use `functorch` [Horace He, 2021] to compute both the gradient and the Hessian-vector-product. We then rely on `scipy` [Virtanen et al., 2020] implementation of the explicit Runge-Kutta method of order $5(4)$ [Dormand and Prince, 1980] to solve the initial-value problem. We use default tolerances in all our experiments.

**Linearized manifold.** We define our linearized manifold by considering the "linearized" function $f_\theta^{\mathrm{lin}}(\mathbf{x}) = f_{\theta_*}(\mathbf{x}) + \langle \nabla_\theta f_\theta(\mathbf{x})\big|_{\theta=\theta_*}, \theta - \theta_*\rangle$ to compute the loss, where $\nabla_\theta f_\theta(\mathbf{x})\big|_{\theta=\theta_*} \in \mathbb{R}^{C\times K}$ is the Jacobian. To compute $\langle \nabla_\theta f_\theta(\mathbf{x})\big|_{\theta=\theta_*}, \theta - \theta_*\rangle$ we use `functorch` to compute a jacobian-vector product. This way, we avoid having to compute and store the Jacobian, which for big networks and large dataset is infeasible to store.

## C.1   Cost of solving the ODE system

Our proposed approach builds on top of Laplace approximation, and therefore it comes with a computational overhead. To be able to estimate this overhead, we have to assume that our solver does $S$ steps to solve an ODE, which is not entirely correct, since the explicit Runge-Kutta method of order $5(4)$ (RK-45) [Dormand and Prince, 1980] uses an adaptive step-size and therefore the value of $S$ differs for every ODE. With this assumption, the cost of evaluating a single ODE results in $O(SWN)$, where $S$ is the number of steps of the solver, $W$ is the number of model parameters and $N$ is the

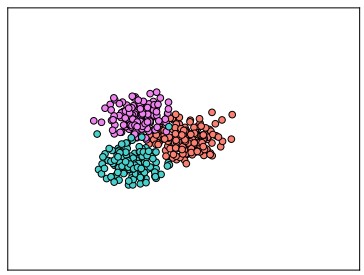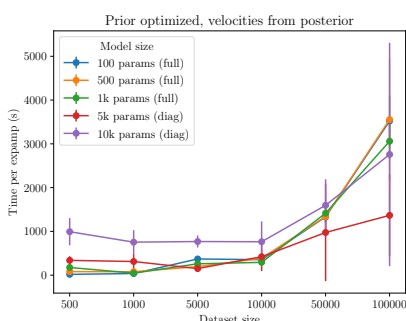

Figure C.1: *Left*: Average runtime per posterior sample in terms of model size and dataset size. *Right*: Average runtime and standard error per posterior sample in terms of model size and dataset size computed over 5 different samples. For the models with $5k$ and $10k$ parameters, we used the diagonal Hessian over all the weights to sample the initial velocities.

dataset size. Indeed, at each step, we need to compute the gradient and the hessian-vector product, which is $O(NW)$, and perform a Runge-Kutta step, which scales linearly with the dimension of the problem, i.e. $O(W)$ [Heirer et al., 1987]. Since the solver would take $S$ steps to solve the system we get $O(SNW)$. If we want to use $K$ posterior samples to estimate our predictive distribution, then we have to solve $K$ different ODEs, therefore the overall overhead is $O(KSNW)$. However, we want to highlight that all these computations happen before deployment of the model. At test time, the complexity is the same as LA when using MC sample to approximate the predictive distribution.

In the derivation of the complexity of our method, we assumed that the ODE solver did exactly $S$ steps. As mentioned before, RK-45 uses an adaptive step-size, therefore the number of steps the solver needs to converge mainly depends on the complexity of the associated ODE problem, which is defined by the geometry of the loss landscape and the initial velocity. To show this, in the next section, we include a benchmark using a synthetic example to analyze how the runtime changes with respect to the size of the model and dataset. While increasing the model parameters and dataset size affect $W$ and $N$, it may be the case that ODE systems do not necessarily get harder, therefore the value of $S$ would be small. In other words, evaluating the ODE becomes more expensive as dimensions increase, but perhaps the actual system gets easier to solve.

**Limitations of our implementation** The cost of obtaining a posterior sample using our RIEM-LA scales linearly in terms of the model and dataset size. In addition to that, our implementation is not highly engineered. Although we can generate these samples in parallel, we use a general purpose Python solver (`scipy.integrate.solve_ivp` [Virtanen et al., 2020]) to solve the ODE system, which runs on the CPU. When the solver needs to evaluate the ODE, our automatic-differentiation based approach runs on the GPU, and the result is moved to the CPU (`.detach().cpu().numpy()`) causing a significant overhead, which is inefficient. Especially when dimensions increase, both the transfer of the data and the computations on the CPU are sub-optimal. Implementing an ODE solver on a suitable automatic-differentiation framework (e.g. JAX) solely running on GPU will dramatically improve performance.

## C.2    Analysis of exponential map

We mentioned scalability as one of the main limitation of our proposed method. We use a simple toy-example (`sklearn.datasets.make_blobs` with three classes, see Fig. C.1 (*left*)) to analyze the running time of computing a single sample from the posterior with our methods. A two layers neural-network is considered, and we vary the number of parameters and datapoints in the dataset. For the biggest models, i.e. the one with $5k$ and $10k$ parameters, we use a diagonal approximation of the Hessian to sample the initial velocities, while for all the others we use the full Hessian. In all cases, the ODE system always relies on the hvp which uses the full Hessian. In Fig. C.1 (*right*), we present runtime for five different models and six dataset sizes. We report the mean and standard deviation computed over five different exponential maps. The results show that the runtime increases

for all models as the dataset size increases. We also see that using a diagonal approximation of the Hessian leads to faster exponential maps.

## D   Experiments details and additional results

### D.1   Regression example

For the regression example we consider two fully connected networks, one with one hidden layer with 15 units and one with two layers with 10 units each, both with `tanh` activations. We train both model using full-dataset GD, using a weight decay of $1e-2$ for the larger model and $1e-3$ for the smaller model for 35000 and 700000 epochs respectively. In both cases, we use a learning rate of $1e-3$. In Sec. 5, we show some samples from the posterior distribution obtained by using vanilla LA and our RIEM-LA approach. In Fig. D.1 we report the predictive distribution for our classic approach while in Fig. D.2 we show both the posterior and the predictive for our linearized manifold and linearized LA. We can see that our linearized approach perform similarly to linearized LA in terms of posterior samples and predictive distribution.

A more interesting experiment is to consider a gap in our dataset to measure the in-between uncertainty. A good behaviour would be to give calibrated uncertainty estimates in between separated regions of observations [Foong et al., 2019]. In our regression example, we consider the points between 1.5 and 3 as test set, and report posterior samples and predictive distribution for both our methods, vanilla and linearized, and LA. From Fig. D.3, we can notice that our RIEM-LA is able to generate uncertainty estimates that are reliable both in the small and the overparametrized model. It gets more interesting when we consider our linearized manifold approach as it can be seen in Fig. D.4. While for a small model the predictive distribution we get is comparable to linearized LA, when we consider the overparametrized model, it tends to overfit to a solution that it is better from the perspective of the linearized loss but very different from the MAP. Therefore, this results in uncertainty estimates that are not able to capture the true data set in this case. This behaviour is related to our discussion in Sec. 3.3. By using a subset of the training set to solve the ODEs system we alleviate this behavior by giving more reliable uncertainty estimates.

For each all the regression setting described above, we also report the mean and the uncertainty estimates by sampling from a closer approximation of the true posterior obtained using Hamiltonian Monte Carlo (HMC) and a No-U-Turn sampler. We use one chain, samples 1000 weights, discard the first 500 as warm-up and use the last 500 to compute the predictive distribution. Results can be seen in Fig. D.5. We can see that in this simple example the uncertainty produced by our method is really close to the one produced by HMC.

### D.2   Illustrative 2D classification example

We consider the `banana` dataset as an illustrative example to study the confidence of our proposed method against Laplace and linearized Laplace. We train a 2-layer fully connected neural net with 16 hidden units per layer and `tanh` activation using SGD for 2500 epochs. We use a learning rate of $1e-3$ and weight-decaay of $1e-2$. Although it is a binary classification problem, we use a network with two outputs and use the cross-entropy loss to train the model because the `Laplace` library does not support binary cross-entropy at the moment. For all methods, we use 100 MC samples for the predictive distribution.

As we have mentioned in Sec. 5, our linearized manifold when we solve the ODEs system by using the entire training set tends to be overconfident compared to the our classic non-linearized approach. This can be easily seen form Fig. D.6 where we can compare the last two rows and see that solving the ODEs system using batches is beneficial in terms of uncertainty quantification. We also plot the confidence of all different methods when we do not optimize the prior precision. We can see that our proposed approaches are robust to it, while linearized LA highly depends on it.

### D.3   An additional illustrative example

We consider the `pinwheel` dataset with five different classes as an additional illustrative classification example. Given the way these classes clusters, it is interesting to see if some of the approaches are able to be confident only in regions where there is data. We generate a dataset considering 200

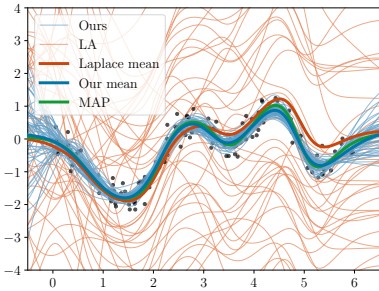
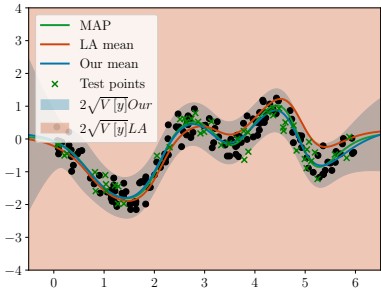

POSTERIOR - SINGLE LAYER MODEL · PREDICTIVE - SINGLE LAYER MODEL

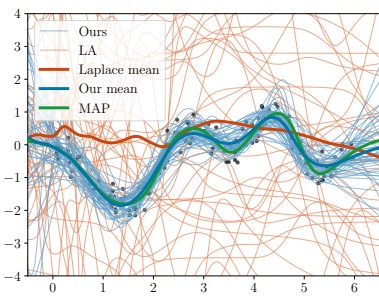
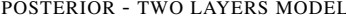
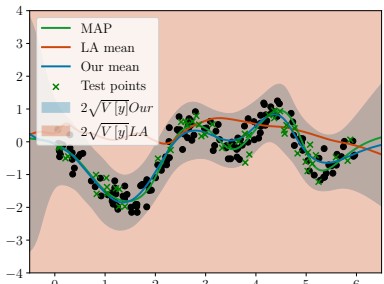

POSTERIOR - TWO LAYERS MODEL · PREDICTIVE - TWO LAYERS MODEL

Figure D.1: Regression results in terms of posterior and predictive distribution using vanilla LA and our RIEM-LA approach. Our proposed approach is able to get better samples and to give a reliable uncertainty estimate compared to vanilla LA. If we consider an overparametrized model.

examples per classes and we use 350 example as training set and the remaining as test set. We consider a two layer fully-connected network with 20 hidden units per layer and `tanh` activation. As before, we train it using SGD with a learning rate of $1e-3$ and a weight decay of $1e-2$ for 5000 epochs.

From Fig. D.7, we can see not only vanilla LA but also linearized LA is failing in being confident also in-data region when we use 50 posterior samples. Our proposed approaches instead give meaningful uncertainty estimates, but we can see that by optimizing the prior precision, our methods get slightly more confident far from the data.

### D.4 Additional results in the UCI classification tasks

In the main paper we present results on five different UCI classification tasks in terms of test negative log-likelihood. Here, we report also results in terms of test accuracy, Brier score, and expected calibration error. Results with prior precision optimized are shown in Table 5, while for prior precision not optimized we refer to Table 6. In both cases, we consider a neural network with a single fully connected layer consisting of 50 hidden units and `tanh` activation. We train it using Adam optimizer [Kingma and Ba, 2015] for 10000 epochs using a learning rate of $1e-3$ and a weight decay of $1e-2$.

From the two tables, we can see that our RIEM-LA is better than all the other methods both in terms of NLL, brier, and ECE. It is also surprising that our method, both the classic and the linearized approach, are able to improve the accuracy of the MAP estimate in most datasets.

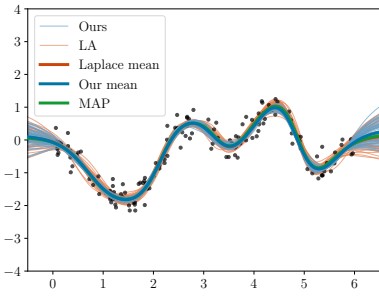
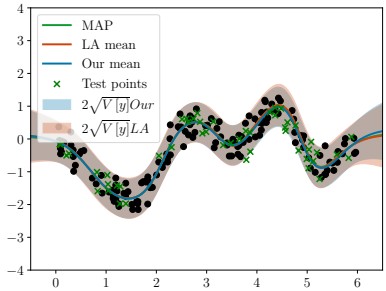

POSTERIOR - SINGLE LAYER MODEL    PREDICTIVE - SINGLE LAYER MODEL

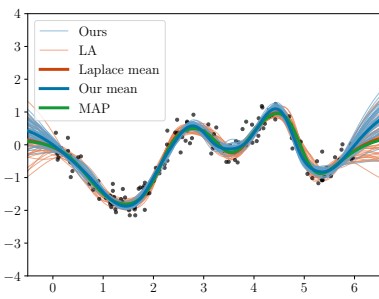
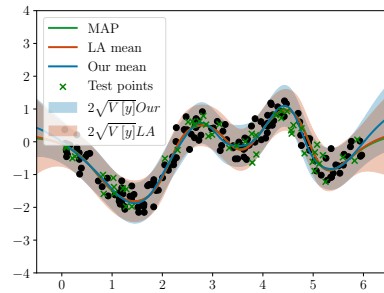

POSTERIOR - TWO LAYERS MODEL    PREDICTIVE - TWO LAYERS MODEL

Figure D.2: Regression results in terms of posterior and predictive distribution using linearized LA and our linearized manifold approach. Our linearized manifold perform similarly to linearized LA no matter what model architecture we are considering.

## D.5 Complete results on MNIST and FMNIST

For these two image classification tasks we consider a small convolutional neural network. Our network consists of the following layers: an initial convolutional layer with 4 channels and $5 \times 5$ filter followed by `tanh` activation and an average pooling layer. The we have another convolutional layer still with 4 channels and $5 \times 5$ kernel also followed by `tanh` activation and an average pooling layer. Then we have three fully connected layer with 16, 10, and 10 hidden units respectively and `tanh` activation.

We train both models using SGD with a learning rate of $1e - 3$ and weight decay of $5e - 4$ for 100 epoch. The learning rate is annealed using the cosine decay method [Loshchilov and Hutter, 2017].

In Table 7 and Table 8, we can see that also when we do not optimize the prior precision our RIEM-LA is mostly performing better than all the alternatives. In particular, for the CNN trained on MNIST and no prior precision optimization, we have that the MAP is also performing well in terms of NLL and Brier score. On FashionMNIST, instead, if we do not optimize the prior we have that both our approaches are better than all the other methods.

## D.6 Out-of-distribution results

The benefit of having meaningful and robust uncertainty estimation is that our model would then be confident in-data region while being uncertain in region without data. Therefore, if this is happening, then we would expect the model to be able to detect out-of-distribution (OOD) examples more successfully.

We consider classic OOD images detection tasks, where we train a model on MNIST and tested on FashionMNIST, EMNIST, and KMNIST and one trained on FMNIST and tested on the remaining datasets. It's well known that linearized LA is one of the strongest method for OOD detection

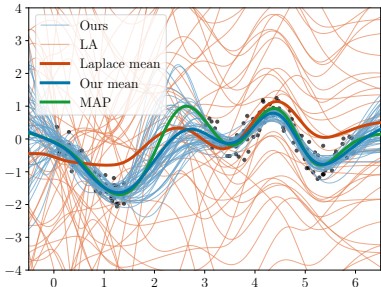

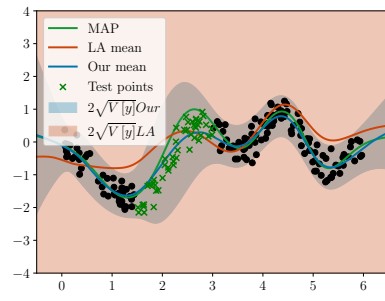

POSTERIOR - SINGLE LAYER MODEL             PREDICTIVE - SINGLE LAYER MODEL

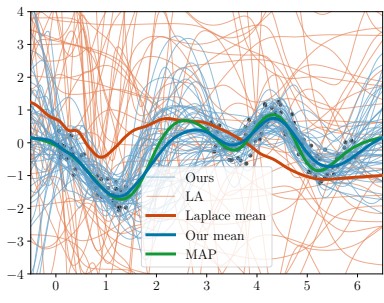

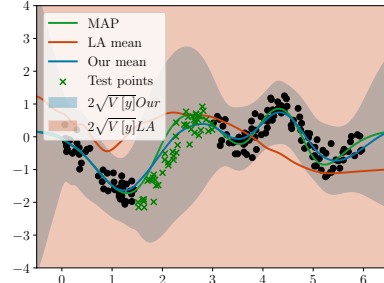

POSTERIOR - TWO LAYERS MODEL               PREDICTIVE - TWO LAYERS MODEL

Figure D.3: In-between uncertainty example using vanilla LA and our `riem-la` approach. Also in this setting, our vanilla approach is able to overcome the difficulties of vanilla LA in this problem setting and giving both meaningful samples from the posterior and a reliable predictive distribution.

[Daxberger et al., 2021a]. We consider the same MAP estimates we used in the previous section and present OOD performance in Table 9 and Table 10.

We can see that our proposed method is consistently working better than linearized and classic LA in all the considered setting apart for models trained on FMNIST where we do not optimize the prior precision to compute our initial velocities. In that setting, however, our linearized approach using batches is getting similar performance than linearized LA.

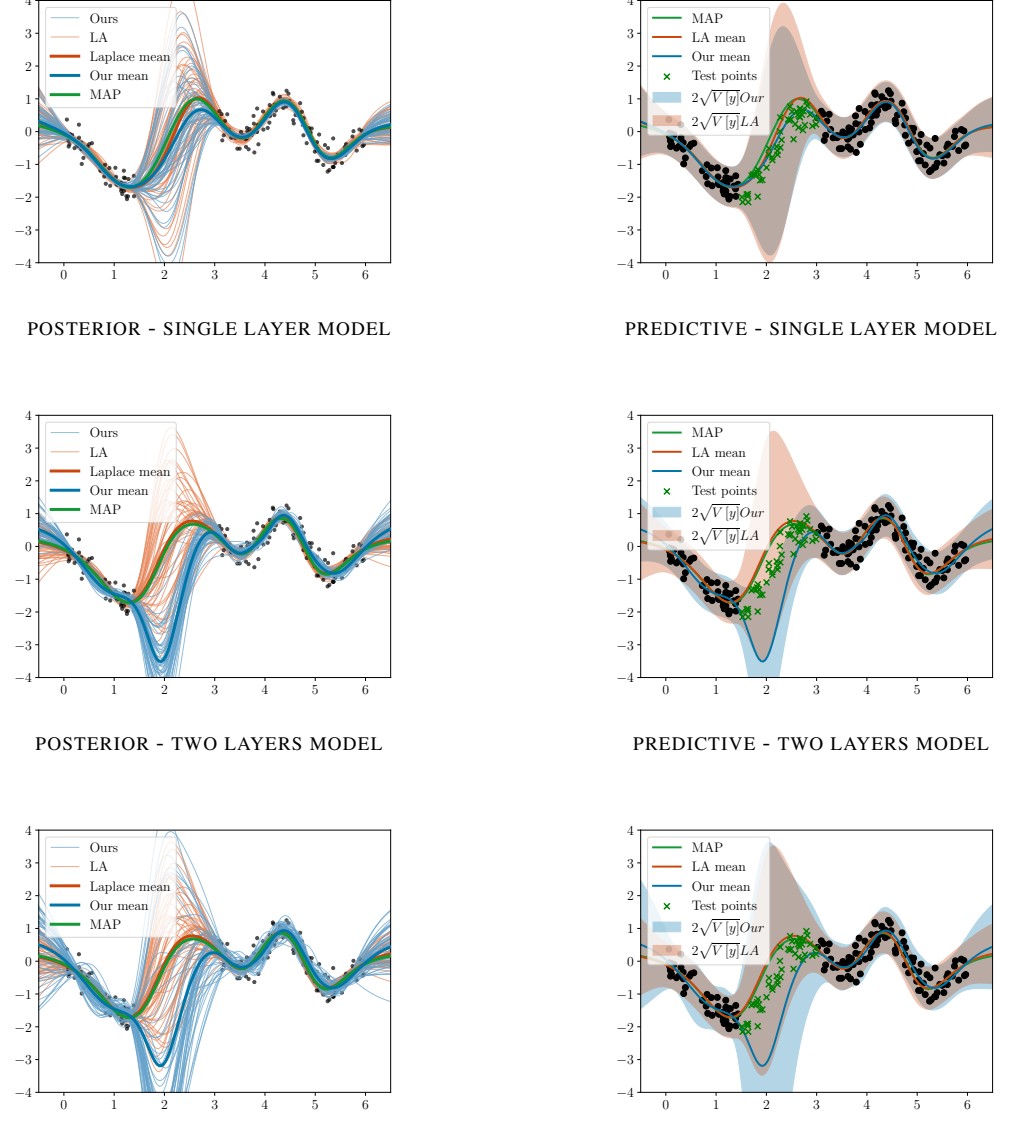

POSTERIOR - SINGLE LAYER MODEL      PREDICTIVE - SINGLE LAYER MODEL

POSTERIOR - TWO LAYERS MODEL      PREDICTIVE - TWO LAYERS MODEL

POSTERIOR - TWO LAYERS MODEL (BATCH)      PREDICTIVE - TWO LAYERS MODEL (BATCH)

Figure D.4: In-between uncertainty example using linearized LA and our linearized manifold. For our linearized approach we both consider solving the ODEs system using the entire training set and using subsets of it. We already highlight how our linearized approach tend to overfit on the linearized loss, and we can clearly see it here from the plots in the center. Using batches to solve the ODEs system gives more reliable uncertainty estimates.

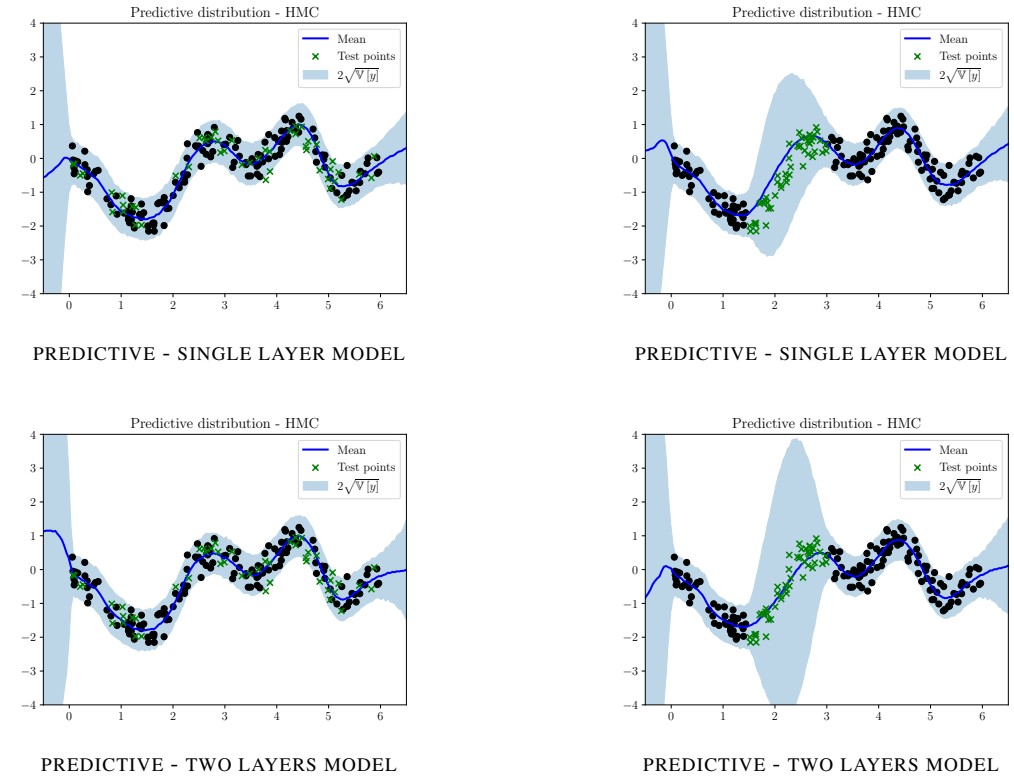

Figure D.5: Predictive distribution obtained using HMC on all the different regression examples considered using a one-layer and a two-layers model.

Table 4: Full in-distribution results in the banana dataset. We used 100 samples both for the Laplace approximation and our method. ECE and MCE are computed using $M = 10$ bins.

| | BANANA DATASET | | | | |
|---|---|---|---|---|---|
| | PRIOR PRECISION NOT OPTIMIZED | | | | |
| METHOD | Accuracy ↑ | NLL↓ | Brier↓ | ECE↓ | MCE↓ |
| MAP | $86.69 \pm 0.34$ | $0.333 \pm 0.005$ | $0.0930 \pm 0.0015$ | $5.91 \pm 0.34$ | $22.75 \pm 2.42$ |
| VANILLA LA | $48.85 \pm 2.32$ | $0.700 \pm 0.003$ | $0.2534 \pm 0.0017$ | $7.10 \pm 1.96$ | $23.10 \pm 6.69$ |
| LIN-LA | $86.92 \pm 0.40$ | $0.403 \pm 0.012$ | $0.1196 \pm 0.0044$ | $13.04 \pm 0.60$ | $17.11 \pm 0.54$ |
| RIEM-LA | $\mathbf{87.14 \pm 0.20}$ | $\mathbf{0.285 \pm 0.001}$ | $\mathbf{0.0878 \pm 0.0006}$ | $\mathbf{2.75 \pm 0.22}$ | $\mathbf{6.30 \pm 0.89}$ |
| RIEM-LA (BATCHES) | $\mathbf{87.32 \pm 0.17}$ | $0.294 \pm 0.002$ | $0.0895 \pm 0.0004$ | $4.79 \pm 0.31$ | $8.28 \pm 0.84$ |
| LIN-RIEM-LA | $85.33 \pm 0.31$ | $0.884 \pm 0.037$ | $0.1252 \pm 0.0022$ | $11.50 \pm 0.31$ | $36.27 \pm 2.84$ |
| LIN-RIEM-LA (BATCHES) | $86.16 \pm 0.21$ | $0.352 \pm 0.002$ | $0.0994 \pm 0.0011$ | $4.06 \pm 0.11$ | $13.67 \pm 0.69$ |

| | PRIOR PRECISION OPTIMIZED | | | | |
|---|---|---|---|---|---|
| METHOD | Accuracy ↑ | NLL↓ | Brier↓ | ECE↓ | MCE↓ |
| MAP | $86.69 \pm 0.34$ | $0.333 \pm 0.005$ | $0.0930 \pm 0.0015$ | $5.91 \pm 0.34$ | $22.75 \pm 2.42$ |
| VANILLA LA | $59.50 \pm 5.07$ | $0.678 \pm 0.009$ | $0.2426 \pm 0.0046$ | $12.08 \pm 2.31$ | $27.49 \pm 5.05$ |
| LIN-LA | $86.99 \pm 0.37$ | $0.325 \pm 0.008$ | $0.0956 \pm 0.0023$ | $6.59 \pm 0.41$ | $11.44 \pm 1.85$ |
| RIEM-LA | $\mathbf{87.57 \pm 0.07}$ | $\mathbf{0.287 \pm 0.002}$ | $\mathbf{0.0886 \pm 0.0006}$ | $\mathbf{2.55 \pm 0.37}$ | $10.95 \pm 1.16$ |
| RIEM-LA (BATCHES) | $87.30 \pm 0.08$ | $\mathbf{0.286 \pm 0.001}$ | $0.0890 \pm 0.0000$ | $2.57 \pm 0.04$ | $\mathbf{6.08 \pm 0.67}$ |
| LIN-RIEM-LA | $87.02 \pm 0.38$ | $0.415 \pm 0.029$ | $0.0967 \pm 0.0024$ | $6.97 \pm 0.43$ | $21.24 \pm 2.28$ |
| LIN-RIEM-LA (BATCHES) | $\mathbf{87.77 \pm 0.24}$ | $0.298 \pm 0.006$ | $\mathbf{0.0887 \pm 0.0011}$ | $\mathbf{2.38 \pm 0.28}$ | $8.04 \pm 1.61$ |

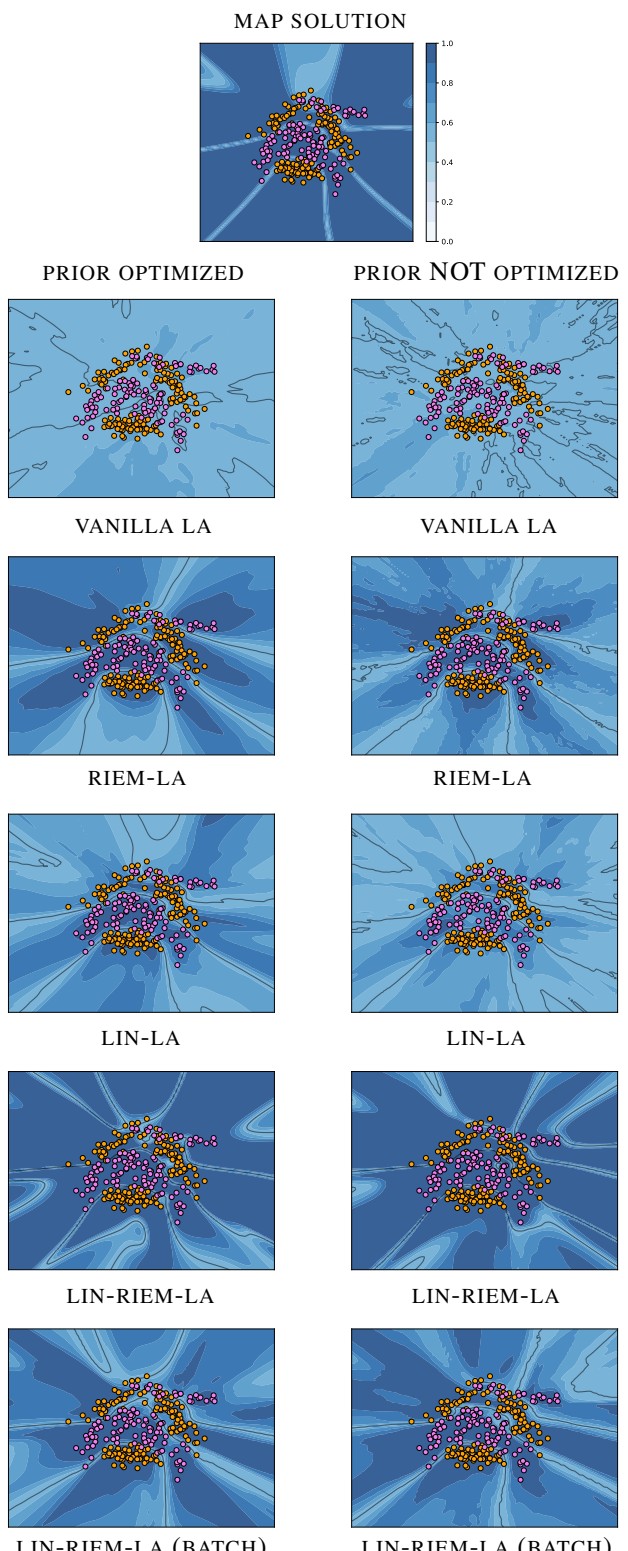

Figure D.6: Confidence for all the considered approach on the `banana` dataset. All plots were computed using 50 MC samples from the posterior. We can notice that our RIEM-LA and LIN-RIEM-LA with batches are giving the best confidence of all methods followed by linearized LA. We can also see that our approaches are robust to prior optimization, while linearized LA gets really better if we optimize the prior precision.

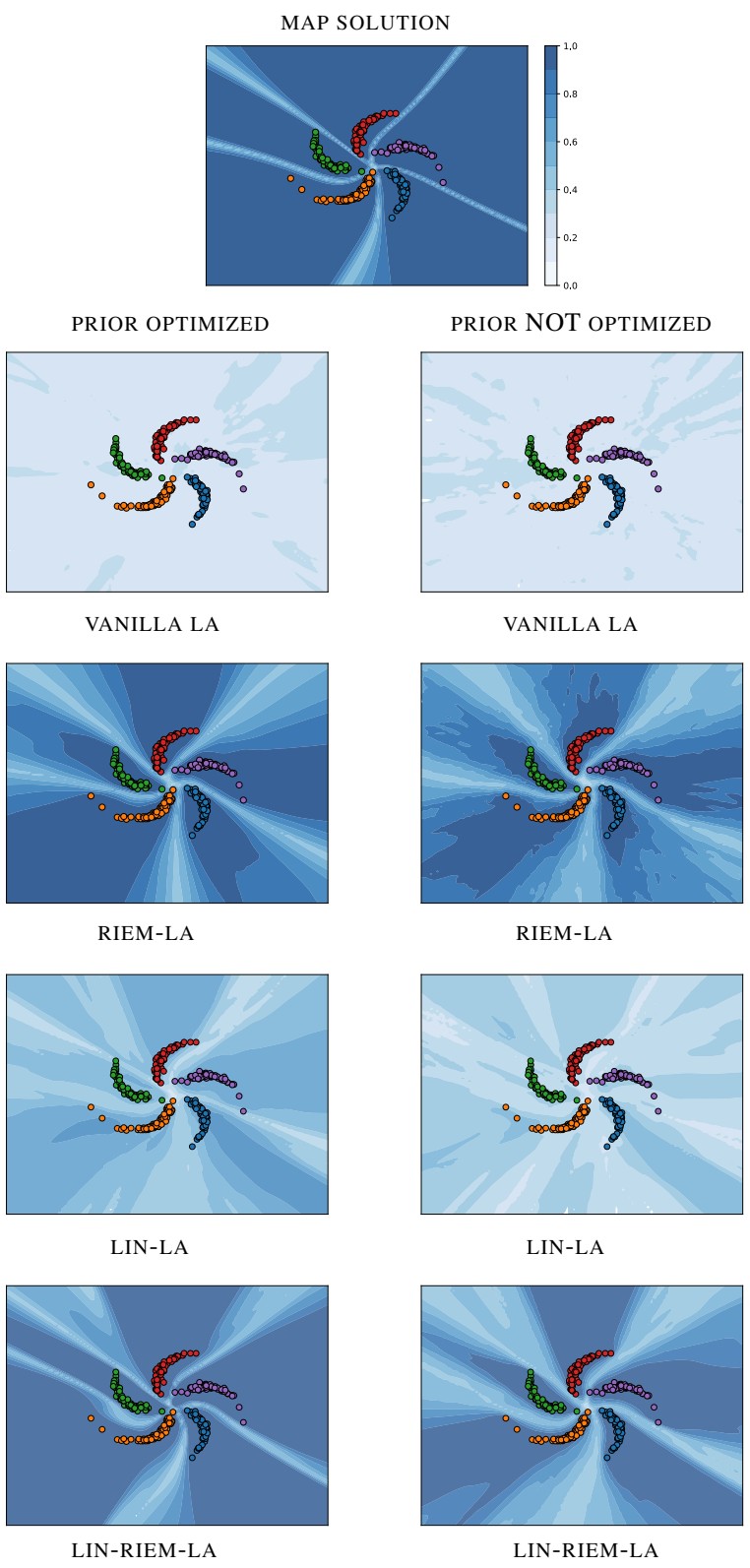

Figure D.7: Confidence for all the considered approach on the `pinwheel` dataset. All plots were computed using 50 MC samples from the posterior. We can notice that in this setting RIEM-LA gives the better confidence followed by LIN-RIEM-LA. However, priorprecision optimization is making our approaches more confident outside the data region. Linearized LA struggle in producing meaningful uncertainty estimates.

Table 5: Results for all the different techniques on UCI datasets for classification. Predictive distribution is estimated using 30 MC samples when prior precision is optimized. Mean and standard error over 5 different seeds.

| | PRIOR PRECISION OPTIMIZED | | | | |
|---|---|---|---|---|---|
| | ACCURACY ↑ | | | | |
| DATASET | MAP | VANILLA LA | RIEM-LA | LINEARIZED LA | LINEARIZED RIEM-LA |
| VEHICLE | **79.52 ± 0.99** | 49.61 ± 3.05 | **80.47 ± 1.12** | 73.86 ± 1.47 | **78.58 ± 1.31** |
| GLASS | 63.75 ± 3.60 | 26.25 ± 3.91 | **67.5 ± 3.40** | 55.62 ± 2.40 | **66.25 ± 3.47** |
| IONOSPHERE | 86.04 ± 2.30 | 58.11 ± 4.04 | **90.19 ± 1.95** | 80.38 ± 1.57 | **88.30 ± 1.12** |
| WAVEFORM | **82.27 ± 1.66** | 64.27 ± 3.19 | **84.93 ± 1.42** | 77.07 ± 1.52 | **82.93 ± 1.08** |
| AUSTRALIAN | 82.30 ± 1.00 | 56.92 ± 2.33 | **84.81 ± 1.03** | 75.00 ± 1.90 | 82.69 ± 0.82 |
| BREAST CANCER | **96.08 ± 0.73** | 71.96 ± 8.58 | **96.86 ± 0.85** | 94.90 ± 0.89 | **96.08 ± 1.30** |

| | NLL ↓ | | | | |
|---|---|---|---|---|---|
| DATASET | MAP | VANILLA LA | RIEM-LA | LINEARIZED LA | LINEARIZED RIEM-LA |
| VEHICLE | 0.975 ± 0.081 | 1.209 ± 0.020 | **0.454 ± 0.024** | 0.875 ± 0.020 | **0.494 ± 0.044** |
| GLASS | 2.084 ± 0.323 | 1.737 ± 0.037 | **1.047 ± 0.224** | 1.365 ± 0.058 | 1.359 ± 0.299 |
| IONOSPHERE | 1.032 ± 0.175 | 0.673 ± 0.013 | **0.344 ± 0.068** | 0.497 ± 0.015 | 0.625 ± 0.110 |
| WAVEFORM | 1.076 ± 0.110 | 0.888 ± 0.030 | **0.459 ± 0.057** | 0.640 ± 0.002 | 0.575 ± 0.065 |
| AUSTRALIAN | 1.306 ± 0.146 | 0.684 ± 0.011 | **0.541 ± 0.053** | **0.570 ± 0.016** | 0.833 ± 0.108 |
| BREAST CANCER | **0.225 ± 0.076** | 0.594 ± 0.030 | **0.176 ± 0.092** | 0.327 ± 0.022 | **0.202 ± 0.073** |

| | BRIER ↓ | | | | |
|---|---|---|---|---|---|
| DATASET | MAP | VANILLA LA | RIEM-LA | LINEARIZED LA | LINEARIZED RIEM-LA |
| VEHICLE | 0.0877 ± 0.0052 | 0.1654 ± 0.0028 | **0.0671 ± 0.0035** | 0.1210 ± 0.0028 | 0.0720 ± 0.0048 |
| GLASS | 0.1058 ± 0.0095 | 0.1353 ± 0.0025 | **0.0740 ± 0.0075** | 0.1102 ± 0.0031 | **0.0787 ± 0.0083** |
| IONOSPHERE | 0.1308 ± 0.0182 | 0.2398 ± 0.0064 | **0.0840 ± 0.0127** | 0.1596 ± 0.0069 | **0.0868 ± 0.0095** |
| WAVEFORM | 0.1043 ± 0.0091 | 0.1766 ± 0.0065 | **0.0825 ± 0.0076** | 0.1260 ± 0.0011 | **0.0871 ± 0.0053** |
| AUSTRALIAN | 0.1642 ± 0.0105 | 0.2452 ± 0.0052 | **0.1209 ± 0.0113** | 0.1901 ± 0.0069 | **0.1340 ± 0.0082** |
| BREAST CANCER | 0.0351 ± 0.0077 | 0.2020 ± 0.0148 | **0.0269 ± 0.0075** | 0.0866 ± 0.0082 | **0.0310 ± 0.0094** |

| | ECE ↓ | | | | |
|---|---|---|---|---|---|
| DATASET | MAP | VANILLA LA | RIEM-LA | LINEARIZED LA | LINEARIZED RIEM-LA |
| VEHICLE | 16.75 ± 1.06 | 15.01 ± 2.41 | **8.43 ± 1.06** | 26.61 ± 1.40 | 10.49 ± 0.95 |
| GLASS | 31.77 ± 2.67 | **11.77 ± 1.08** | 17.06 ± 2.96 | 23.17 ± 1.47 | 20.84 ± 2.57 |
| IONOSPHERE | 13.92 ± 2.02 | 11.22 ± 1.59 | 9.05 ± 0.99 | 15.22 ± 1.38 | **7.70 ± 1.17** |
| WAVEFORM | 15.73 ± 1.29 | 18.73 ± 2.41 | 10.72 ± 1.31 | 19.12 ± 1.60 | **9.10 ± 0.94** |
| AUSTRALIAN | 16.25 ± 0.96 | **5.55 ± 1.41** | 8.49 ± 1.62 | 12.42 ± 1.43 | 10.20 ± 0.90 |
| BREAST CANCER | **3.85 ± 0.89** | 20.39 ± 3.28 | **3.20 ± 0.72** | 20.24 ± 0.99 | **3.16 ± 0.92** |

Table 6: Results for all the different techniques on UCI datasets for classification. Predictive distribution is estimated using 30 MC samples when prior precision is not optimized. Mean and standard error over 5 different seeds.

| | PRIOR PRECISION NOT OPTIMIZED | | | | |
|---|---|---|---|---|---|
| | ACCURACY ↑ | | | | |
| DATASET | MAP | VANILLA LA | RIEM-LA | LINEARIZED LA | LINEARIZED RIEM-LA |
| VEHICLE | $79.52 \pm 0.99$ | $23.94 \pm 1.62$ | $\mathbf{82.05 \pm 1.10}$ | $44.88 \pm 1.74$ | $\mathbf{80.16 \pm 1.45}$ |
| GLASS | $63.75 \pm 3.60$ | $15.00 \pm 3.24$ | $\mathbf{68.13 \pm 3.11}$ | $30.63 \pm 3.89$ | $63.75 \pm 2.88$ |
| IONOSPHERE | $86.04 \pm 2.30$ | $47.55 \pm 1.72$ | $\mathbf{91.32 \pm 1.26}$ | $69.81 \pm 4.10$ | $\mathbf{89.81 \pm 0.68}$ |
| WAVEFORM | $\mathbf{82.27 \pm 1.66}$ | $24.13 \pm 4.30$ | $\mathbf{83.60 \pm 1.34}$ | $54.40 \pm 2.11$ | $\mathbf{83.47 \pm 1.28}$ |
| AUSTRALIAN | $82.30 \pm 1.00$ | $40.19 \pm 1.77$ | $\mathbf{86.73 \pm 1.10}$ | $61.92 \pm 2.57$ | $84.04 \pm 0.80$ |
| BREAST CANCER | $\mathbf{96.08 \pm 0.73}$ | $51.37 \pm 11.68$ | $\mathbf{97.06 \pm 0.73}$ | $84.51 \pm 3.72$ | $\mathbf{95.29 \pm 1.22}$ |

| | NLL ↓ | | | | |
|---|---|---|---|---|---|
| DATASET | MAP | VANILLA LA | RIEM-LA | LINEARIZED LA | LINEARIZED RIEM-LA |
| VEHICLE | $0.975 \pm 0.081$ | $1.424 \pm 0.018$ | $\mathbf{0.398 \pm 0.016}$ | $1.197 \pm 0.018$ | $\mathbf{0.415 \pm 0.016}$ |
| GLASS | $2.084 \pm 0.323$ | $2.057 \pm 0.084$ | $\mathbf{0.981 \pm 0.202}$ | $1.697 \pm 0.044$ | $\mathbf{1.116 \pm 0.270}$ |
| IONOSPHERE | $1.032 \pm 0.175$ | $0.726 \pm 0.011$ | $\mathbf{0.284 \pm 0.080}$ | $0.611 \pm 0.014$ | $\mathbf{0.289 \pm 0.023}$ |
| WAVEFORM | $1.076 \pm 0.110$ | $1.185 \pm 0.031$ | $\mathbf{0.465 \pm 0.051}$ | $0.962 \pm 0.015$ | $\mathbf{0.456 \pm 0.059}$ |
| AUSTRALIAN | $1.306 \pm 0.146$ | $0.750 \pm 0.009$ | $\mathbf{0.518 \pm 0.070}$ | $0.658 \pm 0.012$ | $\mathbf{0.593 \pm 0.030}$ |
| BREAST CANCER | $0.225 \pm 0.076$ | $0.690 \pm 0.051$ | $\mathbf{0.197 \pm 0.093}$ | $0.503 \pm 0.0222$ | $\mathbf{0.171 \pm 0.064}$ |

| | BRIER ↓ | | | | |
|---|---|---|---|---|---|
| DATASET | MAP | VANILLA LA | RIEM-LA | LINEARIZED LA | LINEARIZED RIEM-LA |
| VEHICLE | $0.0877 \pm 0.0052$ | $0.1917 \pm 0.0020$ | $\mathbf{0.0604 \pm 0.0024}$ | $0.1645 \pm 0.0023$ | $0.0657 \pm 0.0027$ |
| GLASS | $0.1058 \pm 0.0095$ | $0.1458 \pm 0.0024$ | $\mathbf{0.0711 \pm 0.0063}$ | $0.1328 \pm 0.0028$ | $\mathbf{0.0768 \pm 0.0067}$ |
| IONOSPHERE | $0.1308 \pm 0.0182$ | $0.2654 \pm 0.0053$ | $\mathbf{0.0689 \pm 0.0076}$ | $0.1596 \pm 0.0069$ | $0.0868 \pm 0.0095$ |
| WAVEFORM | $0.1043 \pm 0.0091$ | $0.2403 \pm 0.0068$ | $\mathbf{0.0789 \pm 0.0056}$ | $0.1929 \pm 0.0033$ | $\mathbf{0.0821 \pm 0.0060}$ |
| AUSTRALIAN | $0.1642 \pm 0.0105$ | $0.2774 \pm 0.0042$ | $\mathbf{0.1081 \pm 0.0096}$ | $0.2328 \pm 0.0059$ | $0.1205 \pm 0.0072$ |
| BREAST CANCER | $\mathbf{0.0351 \pm 0.0077}$ | $0.2486 \pm 0.0248$ | $\mathbf{0.0263 \pm 0.0068}$ | $0.1597 \pm 0.0106$ | $\mathbf{0.0316 \pm 0.0080}$ |

| | ECE ↓ | | | | |
|---|---|---|---|---|---|
| DATASET | MAP | VANILLA LA | RIEM-LA | LINEARIZED LA | LINEARIZED RIEM-LA |
| VEHICLE | $16.75 \pm 1.06$ | $10.78 \pm 1.31$ | $\mathbf{6.65 \pm 0.81}$ | $9.00 \pm 1.32$ | $\mathbf{5.43 \pm 0.74}$ |
| GLASS | $31.77 \pm 2.67$ | $15.27 \pm 1.11$ | $14.84 \pm 2.23$ | $\mathbf{8.18 \pm 2.30}$ | $19.63 \pm 1.74$ |
| IONOSPHERE | $13.92 \pm 2.02$ | $11.77 \pm 2.07$ | $\mathbf{6.32 \pm 0.85}$ | $13.25 \pm 1.99$ | $8.47 \pm 1.27$ |
| WAVEFORM | $15.73 \pm 1.29$ | $18.50 \pm 4.41$ | $\mathbf{6.34 \pm 1.01}$ | $10.17 \pm 1.74$ | $\mathbf{5.32 \pm 0.94}$ |
| AUSTRALIAN | $16.25 \pm 0.96$ | $16.98 \pm 1.70$ | $\mathbf{5.92 \pm 1.50}$ | $\mathbf{7.17 \pm 0.79}$ | $6.36 \pm 0.97$ |
| BREAST CANCER | $\mathbf{3.85 \pm 0.89}$ | $23.00 \pm 5.45$ | $\mathbf{3.27 \pm 0.68}$ | $21.63 \pm 3.01$ | $\mathbf{3.09 \pm 0.55}$ |

Table 7: Results on a simple CNN architecture on MNIST dataset. We train the network on 5000 examples and test the in-distribution performance on the test set, which contains 8000 examples. We used 25 Monte Carlo samples to approximate the predictive distribution and in cases we used a subset of the data to solve the ODEs system we rely on 1000 samples. Calibration metrics are computed using 15 bins.

| | MNIST DATASET | | | | |
| --- | --- | --- | --- | --- | --- |
| | PRIOR PRECISION NOT OPTIMIZED | | | | |
| METHOD | Accuracy ↑ | NLL↓ | Brier↓ | ECE↓ | MCE↓ |
| MAP | $95.02 \pm 0.17$ | $\mathbf{0.167 \pm 0.005}$ | $\mathbf{0.0075 \pm 0.0002}$ | $\mathbf{1.05 \pm 0.14}$ | $39.94 \pm 14.27$ |
| VANILLA LA | $8.10 \pm 0.74$ | $2.521 \pm 0.006$ | $0.0937 \pm 0.0000$ | $11.79 \pm 0.83$ | $35.69 \pm 6.47$ |
| LIN-LA | $94.00 \pm 0.29$ | $0.350 \pm 0.008$ | $0.0143 \pm 0.0004$ | $16.94 \pm 0.20$ | $35.60 \pm 0.09$ |
| RIEM-LA | $\mathbf{96.18 \pm 0.23}$ | $0.177 \pm 0.007$ | $0.0073 \pm 0.0003$ | $7.10 \pm 0.23$ | $\mathbf{25.41 \pm 1.96}$ |
| RIEM-LA (BATCHES) | $94.98 \pm 0.20$ | $0.284 \pm 0.008$ | $0.0111 \pm 0.0004$ | $13.64 \pm 0.20$ | $29.23 \pm 0.50$ |
| LIN-RIEM-LA | $95.12 \pm 0.27$ | $0.180 \pm 0.006$ | $0.0080 \pm 0.0003$ | $4.19 \pm 0.07$ | $\mathbf{22.02 \pm 3.97}$ |
| LIN-RIEM-LA (BATCHES) | $94.63 \pm 0.19$ | $0.229 \pm 0.007$ | $0.0097 \pm 0.0004$ | $7.84 \pm 0.18$ | $28.74 \pm 3.67$ |
| | PRIOR PRECISION OPTIMIZED | | | | |
| METHOD | Accuracy ↑ | NLL↓ | Brier↓ | ECE↓ | MCE↓ |
| MAP | $95.02 \pm 0.17$ | $0.167 \pm 0.005$ | $0.0075 \pm 0.0002$ | $1.05 \pm 0.14$ | $39.94 \pm 14.27$ |
| VANILLA LA | $88.69 \pm 1.84$ | $0.871 \pm 0.026$ | $0.0393 \pm 0.0013$ | $42.11 \pm 1.22$ | $50.52 \pm 1.45$ |
| LIN-LA | $94.91 \pm 0.26$ | $0.204 \pm 0.006$ | $0.0087 \pm 0.0003$ | $6.30 \pm 0.08$ | $39.30 \pm 16.77$ |
| RIEM-LA | $\mathbf{96.74 \pm 0.12}$ | $\mathbf{0.115 \pm 0.003}$ | $\mathbf{0.0052 \pm 0.0002}$ | $2.48 \pm 0.06$ | $38.03 \pm 15.02$ |
| RIEM-LA (BATCHES) | $95.67 \pm 0.19$ | $0.170 \pm 0.005$ | $0.0072 \pm 0.0002$ | $5.40 \pm 0.06$ | $22.40 \pm 0.51$ |
| LIN-RIEM-LA | $95.44 \pm 0.18$ | $0.149 \pm 0.004$ | $0.0068 \pm 0.0003$ | $\mathbf{0.66 \pm 0.03}$ | $39.40 \pm 14.75$ |
| LIN-RIEM-LA (BATCHES) | $95.14 \pm 0.20$ | $0.167 \pm 0.004$ | $0.0076 \pm 0.0002$ | $3.23 \pm 0.04$ | $\mathbf{18.10 \pm 2.50}$ |

Table 8: Results on a simple CNN architecture on FMNIST dataset. We train the network on 5000 examples and test the in-distribution performance on the test set, which contains 8000 examples. We used 25 Monte Carlo samples to approximate the predictive distribution and in cases we used a subset of the data to solve the ODEs system we rely on 1000 samples. Calibration metrics are computed using 15 bins.

| | FMNIST DATASET | | | | |
| --- | --- | --- | --- | --- | --- |
| | PRIOR PRECISION NOT OPTIMIZED | | | | |
| METHOD | Accuracy ↑ | NLL↓ | Brier↓ | ECE↓ | MCE↓ |
| MAP | $79.88 \pm 0.09$ | $0.541 \pm 0.002$ | $0.0276 \pm 0.0000$ | $\mathbf{1.66 \pm 0.07}$ | $24.07 \pm 1.50$ |
| VANILLA LA | $10.16 \pm 0.59$ | $2.548 \pm 0.050$ | $0.0936 \pm 0.0007$ | $10.79 \pm 1.05$ | $27.19 \pm 6.64$ |
| LIN-LA | $79.18 \pm 0.14$ | $0.640 \pm 0.004$ | $0.0308 \pm 0.0001$ | $10.99 \pm 0.54$ | $\mathbf{18.69 \pm 0.57}$ |
| RIEM-LA | $\mathbf{82.70 \pm 0.23}$ | $\mathbf{0.528 \pm 0.004}$ | $\mathbf{0.0262 \pm 0.0002}$ | $9.95 \pm 0.38$ | $19.11 \pm 0.43$ |
| RIEM-LA (BATCHES) | $80.77 \pm 0.10$ | $0.582 \pm 0.004$ | $0.0285 \pm 0.0001$ | $10.34 \pm 0.60$ | $\mathbf{18.52 \pm 0.64}$ |
| LIN-RIEM-LA | $80.94 \pm 0.20$ | $\mathbf{0.528 \pm 0.004}$ | $0.0265 \pm 0.0002$ | $1.59 \pm 0.18$ | $17.58 \pm 3.60$ |
| LIN-RIEM-LA (BATCHES) | $79.95 \pm 0.22$ | $0.567 \pm 0.005$ | $0.0281 \pm 0.0002$ | $4.79 \pm 0.58$ | $\mathbf{16.35 \pm 1.88}$ |
| | PRIOR PRECISION OPTIMIZED | | | | |
| METHOD | Accuracy ↑ | NLL↓ | Brier↓ | ECE↓ | MCE↓ |
| MAP | $79.88 \pm 0.09$ | $0.541 \pm 0.002$ | $0.0276 \pm 0.0000$ | $\mathbf{1.66 \pm 0.07}$ | $24.07 \pm 1.50$ |
| VANILLA LA | $74.88 \pm 0.83$ | $1.026 \pm 0.046$ | $0.0482 \pm 0.0019$ | $31.63 \pm 1.28$ | $43.61 \pm 2.95$ |
| LIN-LA | $79.85 \pm 0.13$ | $0.549 \pm 0.001$ | $0.0278 \pm 0.0000$ | $3.23 \pm 0.44$ | $37.88 \pm 17.98$ |
| RIEM-LA | $\mathbf{83.33 \pm 0.17}$ | $\mathbf{0.472 \pm 0.001}$ | $\mathbf{0.0237 \pm 0.0001}$ | $3.13 \pm 0.48$ | $10.94 \pm 2.11$ |
| RIEM-LA (BATCHES) | $81.65 \pm 0.18$ | $0.525 \pm 0.004$ | $0.0263 \pm 0.0002$ | $5.80 \pm 0.73$ | $35.30 \pm 18.40$ |
| LIN-RIEM-LA | $81.33 \pm 0.10$ | $0.521 \pm 0.004$ | $0.0261 \pm 0.0002$ | $\mathbf{1.59 \pm 0.40}$ | $25.53 \pm 0.10$ |
| LIN-RIEM-LA (BATCHES) | $80.49 \pm 0.13$ | $0.529 \pm 0.003$ | $0.0269 \pm 0.0002$ | $2.10 \pm 0.42$ | $\mathbf{6.14 \pm 1.42}$ |

Table 9: AUROC ↑ for OOD tasks for models trained on MNIST and tested against FashionMNIST, EMNIST, KMNIST. *(B)* indicates exponential maps solved using a subset of the training set. We can notice that our proposed RIEM-LA is consistently performing better than all the other approaches for OOD detection.

| | PRIOR PRECISION NOT optimized | | | | | | |
|---|---|---|---|---|---|---|---|
| OOD DATA | MAP | VANILLA LA | RIEM-LA | LIN. LA | LIN. RIEM-LA | RIEM-LA (B) | LIN. RIEM-LA (B) |
| FMNIST | $0.777 \pm 0.057$ | $0.523 \pm 0.020$ | $\mathbf{0.911 \pm 0.011}$ | $0.876 \pm 0.018$ | $0.873 \pm 0.023$ | $0.891 \pm 0.013$ | $0.862 \pm 0.026$ |
| EMNIST | $0.851 \pm 0.008$ | $0.490 \pm 0.015$ | $\mathbf{0.900 \pm 0.004}$ | $0.897 \pm 0.005$ | $0.905 \pm 0.003$ | $0.872 \pm 0.005$ | $\mathbf{0.901 \pm 0.004}$ |
| KMNIST | $0.877 \pm 0.005$ | $0.511 \pm 0.013$ | $\mathbf{0.949 \pm 0.002}$ | $0.930 \pm 0.002$ | $0.941 \pm 0.002$ | $0.938 \pm 0.003$ | $0.941 \pm 0.002$ |

| | PRIOR PRECISION OPTIMIZED | | | | | | |
|---|---|---|---|---|---|---|---|
| OOD DATA | MAP | VANILLA LA | RIEM-LA | LIN. LA | LIN. RIEM-LA | RIEM-LA (B) | LIN. RIEM-LA (B) |
| FMNIST | $0.777 \pm 0.057$ | $0.681 \pm 0.029$ | $\mathbf{0.917 \pm 0.017}$ | $0.854 \pm 0.024$ | $0.809 \pm 0.027$ | $0.897 \pm 0.012$ | $0.841 \pm 0.025$ |
| EMNIST | $0.851 \pm 0.008$ | $0.768 \pm 0.018$ | $\mathbf{0.917 \pm 0.001}$ | $0.888 \pm 0.006$ | $0.871 \pm 0.003$ | $0.899 \pm 0.003$ | $0.888 \pm 0.003$ |
| KMNIST | $0.877 \pm 0.005$ | $0.813 \pm 0.012$ | $\mathbf{0.953 \pm 0.001}$ | $0.921 \pm 0.003$ | $0.906 \pm 0.002$ | $0.948 \pm 0.002$ | $0.926 \pm 0.002$ |

Table 10: AUROC ↑ for OOD tasks for models trained on FashionMNIST and tested against MNIST, EMNIST, KMNIST. *(B)* indicates exponential maps solved using a subset of the training set. It is interesting to notice that if we do not optimize the prior precision, then LIN-LA is performing the best together with our linearized approaches. If we optimize the prior, RIEM-LA performs the best of all approaches.

| | PRIOR PRECISION NOT optimized | | | | | | |
|---|---|---|---|---|---|---|---|
| OOD DATA | MAP | VANILLA LA | RIEM-LA | LIN. LA | LIN. RIEM-LA | RIEM-LA (B) | LIN. RIEM-LA (B) |
| MNIST | $0.715 \pm 0.022$ | $0.494 \pm 0.014$ | $0.895 \pm 0.010$ | $\mathbf{0.937 \pm 0.012}$ | $\mathbf{0.929 \pm 0.015}$ | $0.917 \pm 0.006$ | $\mathbf{0.940 \pm 0.012}$ |
| EMNIST | $0.649 \pm 0.009$ | $0.495 \pm 0.013$ | $0.794 \pm 0.011$ | $\mathbf{0.907 \pm 0.007}$ | $0.881 \pm 0.009$ | $0.814 \pm 0.016$ | $0.891 \pm 0.008$ |
| KMNIST | $0.718 \pm 0.013$ | $0.495 \pm 0.013$ | $0.886 \pm 0.006$ | $\mathbf{0.924 \pm 0.005}$ | $\mathbf{0.921 \pm 0.007}$ | $0.898 \pm 0.007$ | $\mathbf{0.925 \pm 0.006}$ |

| | PRIOR PRECISION OPTIMIZED | | | | | | |
|---|---|---|---|---|---|---|---|
| OOD DATA | MAP | VANILLA LA | RIEM-LA | LIN. LA | LIN. RIEM-LA | RIEM-LA (B) | LIN. RIEM-LA (B) |
| MNIST | $0.715 \pm 0.022$ | $0.861 \pm 0.037$ | $\mathbf{0.921 \pm 0.011}$ | $0.864 \pm 0.020$ | $0.807 \pm 0.016$ | $\mathbf{0.934 \pm 0.005}$ | $0.869 \pm 0.016$ |
| EMNIST | $0.649 \pm 0.009$ | $0.765 \pm 0.050$ | $\mathbf{0.857 \pm 0.009}$ | $0.806 \pm 0.004$ | $0.750 \pm 0.012$ | $\mathbf{0.851 \pm 0.010}$ | $0.795 \pm 0.001$ |
| KMNIST | $0.718 \pm 0.013$ | $0.827 \pm 0.030$ | $\mathbf{0.910 \pm 0.006}$ | $0.846 \pm 0.010$ | $0.800 \pm 0.009$ | $\mathbf{0.907 \pm 0.006}$ | $0.849 \pm 0.009$ |

