# OpenReview forum: "Riemannian Laplace approximations for Bayesian neural networks"
_NeurIPS.cc/2023/Conference — NeurIPS 2023 poster_

### Official Review · Reviewer_dyBq · 2023-07-03

**Soundness:** 3 good
**Presentation:** 3 good
**Contribution:** 3 good
**Rating:** 7
**Confidence:** 3

**Summary:**

This paper develops a Riemannian Laplace approximation, which is a Laplace approximation that takes into account the Riemannian geometry of the loss surface. The contributions of this paper are as follows: i) showing that such a loss-aware Laplace approximation is better able to capture the true posterior (and predictive); ii) presenting the Riemannian geometry framework for the Laplace approximation, with the Hessian in both the normal and tangential space; iii) a practical algorithm for efficiently integrating the required ODE; iv) experimental evidence on several commonly used datasets.

**Strengths:**

The paper is for the most part very well written, it is for the most part easy to follow (main idea and background). And the authors give several examples (including figures) to help the readers further.
The experimental section is decent, having both a toy example that helps gain intuition and a quantitative evaluation on several standard datasets, where the method is benchmarked against vanilla Laplace approximation and the MAP estimate. I think for the purpose of this paper, the various alternative approximate inference algorithms would not be necessary to compare to, as this is a direct extension of Laplace.

**Weaknesses:**

Even though the paper is generally very well written, I did, however, struggle with some parts of the main section. As a reviewer who is very familiar with Bayesian neural networks and the Laplace approximation but less so with Riemannian geometry, I would prefer to have a more in-depth background section on Riemannian geometry. I think the Laplace approximation section can be shortened if space is needed (e.g. tricks of the trade and strengths and weaknesses could be shortened).

The main weakness is the scalability of the method (see limitations). Solving the ODE takes a very long time. According to Fig. (4), this is in the order of tens of seconds for a mini-batch. Even for very small neural networks. The models used in the experimental section are tiny, e.g. single hidden layer networks with only 50 hidden units. For the CNN, the authors mention that they use 2 conv layers, but I didn't find the kernel size.

It is also not clear to me if and which approximation the authors use for the Hessian. Can the complexity be reduced e.g. with a diagonal Hessian approximation? How would this compare to a diagonal or Kronecker-Factorised approximation of the standard LA?

**Questions:**

In Figure 4, why does the NLL have a minimum and not strictly decrease with larger mini-batch size? Shouldn't the estimate become more and more precise? Is the wall-clock time really on the order of tens of seconds? What is the model size?

For the CNN, what is the kernel size?

Which approximation to the Hessian do you use for the Riemannian Laplace approximation (e.g. Diagonal or Kronecker-Factorized)?

**Limitations:**

The method has been applied to very small neural networks and the authors argue that this is because the method is computationally very expensive. While the authors acknowledge this in the limitations, it potentially limits the applicability to realistic neural network model sizes. It would be important to know more precisely what the computational complexity is (also as a function of the model size). Furthermore, I would like to see a comparison in wall clock time for the entire approximation compared to standard Laplace approximation, not only for different mini-batch sizes, but also for different model sizes, including large model sizes.

---

> ### Author Rebuttal · Authors · 2023-08-09
>
> ## Response to Reviewer dyBq
>
> We would like to thank the reviewer for their positive consideration. We appreciate the time and effort spent in reviewing our paper. We addressed all the remaining concerns below.
>
> **Point 1 / Weaknesses**
> > *“Even though the paper is generally very well written, I did, however, struggle with some parts of the main section. … I think the Laplace approximation section can be shortened if space is needed (e.g. tricks of the trade and strengths and weaknesses could be shortened).”*
>
> Thank you for the feedback. The paper is based on two unrelated fields Bayesian NN and differential geometry. Due to space limitations we included in the main paper the most important information needed for the understanding of the idea, while thorough discussion and further information have been moved in the appendix. We will take into account the feedback and update the camera-ready version accordingly.
>
>
> **Point 2 / Weaknesses and Questions**
> > *"It is also not clear to me if and which approximation the authors use for the Hessian. … How would this compare to a diagonal or Kronecker-Factorised approximation of the standard LA?"*
>
> In all the experiments, we consider the full Hessian for computing the covariance of the Gaussian approximation both for classic and linearized LA and our method. This is computed using the Laplace library which computes the GGN approximation. Therefore, for our method we are sampling the velocities from a Gaussian approximation obtained by computing the full Hessian. In the ODE solver, since we rely on “hpv” of functorch, which also uses the full Hessian in a computationally efficient way, we avoid materializing it.
>
> While for complexity we can refer to the general comment, the influence of the choice of the hessian approximation should be futher investigated, but empirically we have seen that diagonal covariances imply faster geodesics (see benchmark in the attached PDF). We will add also a comparison in the camera-ready appendix, where we will analyse how sampling the initial velocities from less accurate Hessian approximations will affect all models.
>
>
> **Point 3 / Questions**
> > *In Figure 4, why does the NLL have a minimum and not strictly decrease with larger mini-batch size? Shouldn't the estimate become more and more precise? Is the wall-clock time really on the order of tens of seconds? What is the model size?*
>
> Thank you for pointing out this issue, we agree that it can be confusing. In this plot we report the test NLL i.e., the negative log-likelihood on test unseen data. Indeed, the train NLL should follow the behavior you described as the more points in the batch, the closer the sampled functions to the MAP estimate. However, the MAP may not be the optimal model for the test distribution. Instead, using a (small) batch when solving an ODE system allows our method to generate functions similar to the MAP, but which exhibit some variability and potentially generalize better. Indeed, if the batch is still representative of the whole dataset, the sample will differ from the MAP mostly close to the decision boundary and away from data.
> Regarding the implementation, the result corresponds to the Banana experiment in Sec 5.2. We will update the plot and the description accordingly.
>
>
> **Point 4  / Questions**
> > *“For the CNN, what is the kernel size?”*
>
> We used a $5\times5$ kernel, as reported in Appendix D.5

---

> > ### Comment · Reviewer_dyBq · 2023-08-16
> > **Review score update**
> >
> > Thank you for your response, addressing my main concerns and questions.
> > I decided to increase my score from 6 to 7.

---

### Official Review · Reviewer_BssD · 2023-07-04

**Soundness:** 3 good
**Presentation:** 4 excellent
**Contribution:** 3 good
**Rating:** 6
**Confidence:** 4

**Summary:**

The paper presents a Laplace approximation for Bayesian neural networks that adapts the covariance to the local geometry of the loss, effectively overcoming the quadratic approximation of the loss. The authors report competitive performance with the standard Laplace approximation (both Monte Carlo sampled and linearized) on regression and MNIST-scale classification problems as well as a reduced reliance on tuning the precision of the prior.

The approach is explained clearly and makes a lot of sense (at least with my rather superficial understanding of Riemannian geometry), it is applicable in more general probabilistic models and I would expect it to lead to various follow-up works. While methodologically this is a very nice paper, I feel like it is let down by the empirical evaluation. The method is only tested on UCI and MNIST-scale datasets, which are hardly relevant for deep learning these days. The authors mention the computational cost of their method, but only discuss the reasons superficially without providing exact benchmark figures to give a sense of where the main bottlenecks arise. Given the apparent computational limitations of the method, an experiment with a non-NN model could have strengthened the paper.

All things considered, the strong methodological contribution outweighs the unconvincing empirical evaluation for me, so I would lean towards acceptance, although I wish could have given the paper a much higher rating.

**Strengths:**

* The core idea makes a lot of sense, is applicable beyond inference in neural networks and seems to work well for the experiments that are considered. It has the potential to address the rather restrictive approximation of a quadratic loss in the Laplace approximation.
* I am confident that the paper will inspire various pieces of follow-up work.
* The paper is well-structured and -written.
* Effective use of illustrative examples throughout.

**Weaknesses:**

* Only small scale problems are considered in the experiments
* This is exacerbated by lack of analysis of the computational cost. It is not really clear to me what specifically is preventing the method from being applied to larger networks and datasets (even something like CIFAR with ResNets would have been great). The discussion mentions scaling issues w.r.t. number of datapoints and parameters, but the relative behavior is not benchmarked at all and it is also not clear to me how much time the ODE solver actually spends e.g. calculating the Hessian-vector products vs computations independent of that. Given that this paper opens up a new direction for research, I think there should be much clearer pointers as to where exactly the current bottlenecks and limitations are and where improvements can realistically be achieved.
* There is no experiment demonstrating the efficacy of the approach on a non-NN probabilistic model. Given the apparent computational cost of the method, this would have seemed like a rather natural experiment to include and neural networks don’t seem like a particularly good fit for the approach.
* The explanation of why the method would work better with a mini-batch rather than the full dataset (section 3.3/fig 4) is not exactly clear and seems rather hand-wavy.
* Lack of HMC ground-truth baselines for the regression problems

**Questions:**

* What are the absolute runtimes of each method per sampled prediction in the expeirments? It would be great to get a better sense of this as a function of dataset/minibatch size and number of parameters/network size (I could see a synthetic experiment be illuminating here).
* How much of the cost of the method lies in calculating the Hessian-vector products? These might take up a significant chunk of the total ODE compute time, so I’m wondering if explicit Hessian approximations (last-layer, KFAC, subset, ...) might improve scalability?
* Could you elaborate on the discussion at the end of section 3.3? In particular I don’t follow how using the full dataset would over-regularize the geodesic as judging from the $N/B$ factor in the inline equation you seem to correctly rescale the mini-batch loss to match the full loss in expectation (note: I assume by ‘over-regularize’ you mean concentrating the samples around the mean, i.e. effectively reduce the entropy of the Gaussian approximate posterior). Any ideas for overcoming this over-regularization?
* Could you comment on the results for the Riemannian Laplace approximation being noticeably better on the UCI datasets with a non-optimized precision?


**Minor note**: the x axis labels in Fig 4 are cut off at the bottom

**Limitations:**

Only discussed superficially and in a quite hand-wavy manner, I would have wanted to see concrete benchmarks and an analysis of how the compute time evolves with increasing dataset size and number of parameters respectively as well as some evidence from the literature that a tailor-made ODE solver could indeed allow the method to take the step from small conv nets to more modern architectures.

---

> ### Author Rebuttal · Authors · 2023-08-09
>
> ## Response to Reviewer BssD
>
> We thank the reviewer for the positive consideration of our work and constructive feedback. We appreciate the time spent to review our paper. We address all your concerns below and we refer to the general comment for the questions regarding scalability.
>
>
> **Point 1/ Weaknesses**
> > *“There is no experiment demonstrating the efficacy of the approach on a non-NN probabilistic model. … good fit for the approach.”*
>
> Indeed, our model can be also used to approximate posteriors other than BNNs. We have already in the paper two constructive examples: the Rosenbrock function (Fig. 1) and the logistic regression (Fig. 3). We are happy to include more examples in the camera-ready appendix. The reasons why we decided to focus on the BNN problem are due to the challenges, the potential impact and the research questions that arise. Even with the current basic setting, our approach performs on-par/better than linearized LA, which is considered among the strongest approximations for turning NNs into BNNs. We also note that our method is interpretable compared to linearized LA, which performance is not yet theoretically understood.
>
>
> **Point 2/ Weaknesses**
> > *"The explanation of why the method would work better with a mini-batch rather than the full dataset seems rather hand-wavy"*
>
> We agree that we have not yet fully analyzed the influence of batching on the result. The terminology can lead to misunderstanding, but for “batching” in this setting we refer to solving each ODE only using a subset of the data. In the context of this paper we propose this as an “obvious and simple” way to scale the method. However, further research should be conducted to properly analyze the behavior. We believe that this is closely related to the generalization concept in deep learning, when stochasticity is induced in the training algorithm.
>
> **Point 3/ Weaknesses**
> > *"Lack of HMC ground-truth baselines for the regression problems"*
>
> Thank you for the suggestion. We added the predictive distribution obtained using HMC on the attached pdf. We will add them in the appendix too.
>
>
> **Point 4 / Questions**
> > *"What are the absolute runtimes of each method per sampled prediction in the expeirments? ...(I could see a synthetic experiment be illuminating here)"*
>
> Thank you for the suggestion. We provide some initial results in the attached PDF, and we will include further analysis in the camera-ready version. We also briefly mention the challenges that influence this benchmark on the general comment.
>
>
> **Point 5 / Questions**
> >*“How much of the cost of the method lies in calculating the Hessian-vector products? … might improve scalability?*
>
> This is a great suggestion for future research. Indeed approximations to the metric and/or the ODE are of particular interest to reduce complexity, as we also mentioned in the general comment.
>
> **Point 6 / Questions**
> > *"Could you elaborate on the discussion at the end of section 3.3? ... Any ideas for overcoming this over-regularization?"*
>
> Indeed, the over-regularization means that due to the linearization and the prior precision optimization, especially if precision gets a high value, the low loss region concentrates closely around the MAP. Therefore, our samples are generated only near the MAP, and this bias limits the variability of the sampled functions. This is for example not happening in our standard approach. We believe that the batching is an interesting way to alleviate this issue as it seems empirically to be beneficial. However, it poses challenging questions and perhaps future insights. For example, there might be a correlation between the quality of the MAP with respect to generalization, and the sampled models associated with the loss surfaces implied by each batch.
>
>
>
> **Point 7 / Questions**
> > *“Could you comment on the results for the Riemannian Laplace approximation being noticeably better on the UCI datasets with a non-optimized precision?”*
>
> If the optimized prior precision is high, this corresponds to a stronger L2 regularization, which implies that more models will be generated near the MAP, and this is true even for our model. When the prior is not optimized, our model is capable of generating models with higher variability, which helps to calibrate the uncertainty better. Therefore, the test NLL is expected to be better in our case, as points near the boundary that are missclassified do not get high confidence. Instead, this is the case when the prior is optimized, since the sampled functions are closer to the MAP.

---

> > ### Comment · Reviewer_BssD · 2023-08-15
> > **Response**
> >
> > Thank you for your comments and providing runtime results and HMC references. I would still really love to see a quantitative non-NN comparison to the regular Laplace approximation. I unfortunately do not have a specific one in mind to suggest, but would have a look through the tutorials/examples of a couple of probabilistic programming frameworks that focus on MCMC for inference. I'm sure they will have comparisons where sampling works much better than VI with a Gaussian/Laplace, and it would be interesting to see to what extent using your Riemannian approach to adapt to the posterior closes the gap (assuming it does). I would also be curious how the method compares e.g. to normalizing flows in such a lower dimensional case. I understand the temptation of wanting to do neural nets first and foremost, but I think there is a really clear path for potential applications with more traditional probabilistic models, whereas BNNs will require more work on scalability. Both are interesting in terms of research of course, but for the impact of the paper it would, at least in my opinion, make a lot of sense to cover the former empirically.
> >
> > Overall and in light of the other reviews with there being a consensus for acceptance, I remain with my score.

---

> > > ### Author Response · Authors · 2023-08-19
> > > **Response to Reviewer**
> > >
> > > Thanks again for your comments and suggestions. We agree with you that a quantitative non-NN comparison to the regular Laplace approximation would be interesting. We also agree that exploring how our approach performs in more traditional probabilistic models instead of BNNs would be interesting to cover. While we are currently looking for examples in the literature to test the latter, we have conducted preliminary experiments in the 2D Rosenbrock density. Following [1], where they define how to get sample from that density, we measured the Wasserstein distance between HMC, LA, and our approach from the true samples. Results are in the table below and they are commputed using 5000 samples.
> > >
> > > | Method               | Wasserstein distance |
> > > | --------             | -------              |
> > > | HMC                  | 7.189                |
> > > | Our                  | 8.398                |
> > > | LA                   | 31.194               |
> > >
> > > [1] Pagani, F., Wiegand, M., & Nadarajah, S. (2019). An n-dimensional rosenbrock distribution for mcmc testing. arXiv preprint arXiv:1903.09556.

---

### Official Review · Reviewer_am4X · 2023-07-06

**Soundness:** 3 good
**Presentation:** 3 good
**Contribution:** 3 good
**Rating:** 6
**Confidence:** 2

**Summary:**

This paper presents a novel Laplace Approximation for Bayesian Neural Networks. A key insight is to examine the local loss landscape with a Riemannian metric, which is determined by the gradient of the log posterior. Using this metric and an exponential map, a Laplace Approximation technique is developed to draw posterior samples that fall into regimes with low negative log posterior. The paper also develops a sampling method, which relies on the 2nd order ODE solver. Several experiments are conducted. When compared to the standard Laplace Approximation, evidences are provided to illustrate the improvements.

**Strengths:**

- the contribution provided by this work is original and novel to the best of my knowledge.

- the paper is polished well generally. Despite that the materials are developed on differential geometry, intuitions are relatively provided well.

- Laplace Approximation has been increasingly popular in recent years. Such extensions to incorporate Riemannian geometry could be relevant to the Bayesian Deep Learning community.

**Weaknesses:**

One complaint about the paper is that, without referring to the appendix, it is difficult to comprehend the material fully. For example, in section 3.3, I wish that the connection between an ODE and Riemannian metric is difficult to understand directly. Differential geometry is not often thought in engineering courses at many universities. It may make sense to recap the essential concept in the main paper. How the method could be used for linearized Laplace Approximation is made very short.

Another point for improvement is the choice of the baselines. It would make sense to include a deep ensemble and MC-dropout as a minimum. This could show how far the proposed Laplace Approximation can compete with popular methods in practice.

In the experiments, the paper could improve on analyzing the computational complexity, in comparison to the standard Laplace Approximation. DaxBerger et al 2021 claim that the major benefit of Laplace Approximation is on simplicity, it would be great to project this paper's method more on the plateau of quality of uncertainty Vs computational complexity. While it is great for research, I think the methods based on differential geometry may have certain drawbacks among practitioners. The paper could be more convincing by providing the gains in uncertainty, but additional overhead due to the added complexity of the pipeline.

**Questions:**

N/A

**Limitations:**

There is a limitation section at the end of the paper.

---

> ### Author Rebuttal · Authors · 2023-08-09
>
> ## Response to Reviewer am4X
>
> We would like to thank the reviewer for their thoughtful consideration of our work. We appreciate the time you took to review our paper. We have taken the time to address all the points raised under Weaknesses.
>
> **Point 1/ Weaknesses**
> > *"...without referring to the appendix, it is difficult to comprehend the material fully."*
> > *"...How the method could be used for linearized Laplace Approximation is made very short."*
>
> Thank you for the feedback. The paper is based on two unrelated fields Bayesian NNs and differential geometry. Due to space limitations thorough discussion and further information have been moved in the appendix, as well as a concrete example for the linearized version of our method. As our standard approach usually works better than the linearized version and is interpretable, we mainly focus on that in the paper. However, we will take into account the feedback and update the camera-ready version accordingly.
>
> **Point 2/ Weaknesses**
> > *"Another point for improvement is the choice of the baselines. …  compete with popular methods in practice"*
>
> In related works where extensions of LA are proposed, linearized LA is considered as the main baseline for comparison, as this is the LA approach that competes well with Deep-Ensemble for Bayesian NNs. Therefore, it is already a quite strong baseline. We are happy to include more comparisons (ensemble methods and dropout) to the camera-ready version.
>
> **Point 3/ Weaknesses**
>
> > *"In the experiments, the paper could improve on analyzing the computational complexity, in comparison to the standard Laplace Approximation. ... due to the added complexity of the pipeline."*
>
> LA is already simple and cheap. Our method is more flexible and expressive, but it comes with the price of increased computational cost as all differential geometric techniques. However, implementing our method is simple as in practice only an ODE (initial value problem) has to be solved when generating a sample (for scalability and complexity see the general comment). Another benefit of our method is interpretability. Even if linearized LA works well, it is not yet understood theoretically why it performs as such. Instead, our method is interpretable, which can be easily seen from the regression example where the LA is known to behave poorly.
>
>
> We mentioned in the general comment that the computational cost on top of Laplace of our proposed method is $O(SNW)$ where $S$ is the number of step of the ODE solver, $N$ is the number of datapoints and $W$ is the number of parameters in the model. Getting $K$ samples from the posterior is therefore $O(KSNW)$. To briefly explain the complexity cost, at each step of the solver, we need to compute the gradient and the hessian-vector product, which are $O(NW)$, and perform a Runge-Kutta step, which scales linearly with the dimension of the problem, i.e. $O(W)$ [1]. Since the solver would take $S$ steps (RK-45 uses an adaptive step-size therefore this values changes in every ODE), we get $O(SNW)$.
>
> 1. Hairer, E., Nørsett, S. P., and Wanner, G. Solving Ordinary Differential Equations I, Nonstiff Problems. Springer, 1993.

---

> > ### Comment · Reviewer_am4X · 2023-08-16
> > **Response to the Authors**
> >
> > I would like to thank the authors for the efforts. I have read other reviews as well as related responses.
> >
> > I stand by the current score -- I think clearly analyzing the computational complexity vs empirical gains in performance (my third point) is one missing point in the paper. This is also connected to some of the raised concerns by other reviewers. While authors discuss the computational complexity, it might help to include empirical results, especially for all the comparison results with standard laplace approximation. If the paper gets accepted, I also hope to see more baselines like MC dropout and deep ensembles.

---

> > > ### Author Response · Authors · 2023-08-16
> > > **Clarification on suggested empirical study**
> > >
> > > Thank you again for the useful feedback and comments, and for engaging in the discussion. We are really keen in adding this additional analysis and providing results before the end of the discussion period. However, we first need some clarification on what kind of experiment the reviewer would like us to perform in order to measure the computational complexity vs empirical gains in performance. If we take the test NLL for example, that measures already the quality of the produced uncertainty. Would you like us to also measure the time it takes to get the posterior samples using our method against Laplace? In addition to the complexity, we evaluate empirically the runtime of generating a sample from different model sizes and different dataset sizes (see Fig. 1.b in the PDF attached to the general comment).
> > >
> > > We would appreciate your clarification so that we can proceed with the additional analysis as soon as possible. Thank you again for your time and consideration.

---

### Official Review · Reviewer_CqAb · 2023-07-07

**Soundness:** 3 good
**Presentation:** 3 good
**Contribution:** 3 good
**Rating:** 5
**Confidence:** 3

**Summary:**

The Laplace approximation offers a practical posterior but is limited due to the symmetry of the weight space it is parameterised in. The method proposed to improve posterior quality by adapting the posterior shape through a Riemannian metric that is determined by the log-posterior gradient.

**Strengths:**

* Practically

The community is interested in using Laplace approximations for Bayesian posterior estimation due to its simplicity and wide applicability. The proposed paper deals with improving the quality of such approximations, which would benefit many.

* Methodologically

Since the quality of the Laplace approximation is inherent to the space it is parameterised in, it is very interesting to see how Riemmanian geometry of the loss landscapes provides a better understanding of the approximations and provides avenues to improve posterior quality.

* Results

It is very promising to see that the Riemannian Laplace approximation allows for good posterior fits even when no linearization or prior tuning is being used.

**Weaknesses:**

Overall, I think the promise of the paper of using the Riemmannian geometry of the loss surface to improve the posterior quality of Laplace approximation is very promising. My concerns mainly lie in the practicality of the approach in terms of scaling the method to larger models (deeper networks and models with more parameters) and larger datasets. Since the paper presents the method in the context of Bayesian deep learning, the scalability of the method is important.

- Scalability to deeper networks
The posterior samples from 'POSTERIOR - TWO LAYERS MODEL' of Fig. D.4 seem off. It would be good if the authors addresses how the method would perform for more complex model classes. Does the method break down in this case, or could this potentially be mitigated?

- Scalability to models with more parameters

As mentioned in the paper, there is a computational cost associated with the growing dimensionality of the parameter space because the number of necessary solver steps increases. I am worried that the method can not be applied to larger deeper NNs as the solutions that are found by ODE solver in practice for larger dimensions will not be of sufficient quality. Since deep neural networks typically consist of many more parameters than the models, this seems like a very big limitation.

- Scalability to larger datasets

The linear scaling in the number of data points.

- Quantitative results and comparisons

The method only considers very small models (e.g. single or 2 layer NNs) and small toy problem datasets. This small data regime would allow computing of (close to) exact posteriors, which would allow better quantitative assessment of the posterior found by Riemmanian Laplace approximation. Furthermore, it would be interesting to see larger model and data regimes.

**Questions:**

a) What are the method's memory and computational budget limitations?

b) How accurate is the ODE solver when the number of dimensions grows?
If the quality of the solutions degrades in higher dimensions, can we expect the method to remain functional for larger models?

c) What are current limitations to scale the method to larger models and datasets (e.g. resnets/transformers models at cifar/imagenet scale)?

d) MacKay also considered notes that the choice of basis for a Laplace approximation is important in [1], which might be a relevant reference. What are the most important reasons to consider the proposed adaptation using the Riemannian metric over other potential changes of the basis?

[1] MacKay, David JC. "Choice of basis for Laplace approximation." Machine learning 33 (1998): 77-86.



**Limitations:**

As mentioned, I think being able to adapt the quality of Laplace approximations by considering geometrical aspects of the loss landscape is very interesting. My concerns are mainly in practicality of the approach.

---

> ### Author Rebuttal · Authors · 2023-08-09
>
> ## Response to Reviewer CqAb
> We thank the reviewer for the positive consideration on our work. We also appreciate the time taken to review our paper. We addressed all concerns you highlight under the Weaknesses and Questions sections.
>
>
> **Point 1 / Weaknesses**
> > *... Fig. D.4 seem off. It would be good if the authors addresses how the method would perform for more complex model classes. Does the method break down in this case, or could this potentially be mitigated?*
>
> In this example, we consider the linearized version of our approach. The data that we use imply a specific loss landscape structure that our samples respect. In the regions where data exist our samples agree with the MAP, while in the other regions the behavior of the samples cannot really be "judged" but potentially the uncertainty. Interestingly, if the prior is not optimized, then our method behaves "as expected" respecting the MAP even in the region without the data (see attached PDF for a plot in that setting). The reason is that with the prior optimized the loss landscape of our linearzied approach has a particular "biased" behavior, which batching seems to help alleviating.
>
> **Point 2 / Weaknesses**
> > *...This small data regime would allow computing of (close to) exact posteriors, which would allow better quantitative assessment...*
>
> Thank you for the suggestion. In the attached PDF we added a comparison with the predictive distribution obtained by using HMC on the regression examples.
>
>
> **Point 3 / Weaknesses**
> > *The method only considers very small models (e.g. single or 2 layer NNs) and small toy problem datasets.*
>
> For the experiments we use an off-the-shelf solver for solving ODEs in ~5000 dimensions. We also use it for a LeNet with ~44000 parameters (see attached PDF). We would like to remark that from a differential geometry viewpoint this is already a surprising result. We expect to be able to push further the dimensions by developing suitable ODE solvers.
>
>
> **Point 4 / Questions**
> > *What are the method's memory and computational budget limitations?*
>
> As we explained in the general comment the overhead our method has on top of Laplace approximation is given by the need to solve an ODE to get a sample. If we define $N$ to be the number of datapoints, $W$ to be the number of parameters, then both computing the gradient and the hvp is $O(NW)$. A single step of Runge-Kutta method of order $5(4)$ is $O(W)$ [1]. At each step we have to evaluate the ODE which requires the computation of gradient and hvp. Therefore, if we assume that the solver performs $S$ steps, the computational cost of the solver is $O(SNW)$.
>
> 1. Hairer, E., Nørsett, S. P., and Wanner, G. Solving Ordinary Differential Equations I, Nonstiff Problems. Springer, 1993.
>
>
> **Point 5 / Questions**
> > *How accurate is the ODE solver when the number of dimensions grows?*
>
> The accuracy of the solution does not depend on the dimensions of the problem per se, but on the complexity of the ODE problem and the solver. We solve ODEs using a general purpose Numpy ODE solver that is based on a high-accuracy algorithm (Runge-Kutta), where accuracy here means that the solution satisfies perfectly the ODE system for all time steps. Therefore, our solutions are highly accurate, with the price of increased computational complexity. We refer to the general comment for potential improvements in efficiency. We also conjecture that “this type of accuracy” may not be critical for our method. Instead, it may be sufficient and "accurate" for our method the generated geodesic to travel within low loss region and stop when the loss increases. A specialized solver with this characteristics is of particular interest.
>
> **Point 6 / Questions**
> > *What are the most important reasons to consider the proposed adaptation using the Riemannian metric over MacKay's choice of basis for a Laplace approximation?*
>
> Thank you for the reference, this is indeed related to our work and we will include it the associated section. MacKay proposed to reparametrize the model/parameter space such that to be as near as possible to a Gaussian. This way, LA will be a good approximation. However, this is not always straightforward, especially in the deep networks regime. Our method is “similar” in spirit, but instead of reparametrizing the parameter space we make the approximation adapt to it by finding several local bases instead of a general one.

---

### Author Rebuttal · Authors · 2023-08-09

## General Comment to all reviewers
We would like to thank the reviewers for their thoughtful comments, positive considerations and suggestions for improving the paper. We appreciate that you found our work novel, well-written, with potential impact for the community, and inspiring for follow-up works. We make a first general comment about the scalability of our approach and we also reply individually to each reviewer. We are happy to clarify further during the discussion phase if some concerns remain.

**Scalability**

The scalability of our proposed approach and the applicability to big models was a common topic of discussion among the reviewers, which they acknowledge that we also highlight in the paper as the main limitation of our approach. We will elaborate more about this issue in the camera-ready version based on the following discussion.

**Evaluation of the ODE**:
- Based on the structure of the Riemannian metric we simplified the original ODE, and its final form allows to apply automatic-differentiation to evaluate it. Otherwise, evaluating the original ODE is rather prohibitive as it needs to compute the Hessian and the Gradient of the loss individually.
- Evaluating the ODE needs all the training data points which is prohibited with big data. We proposed the “obvious” trick of using a random (small) batch when solving an ODE, which empirically in some cases even boosts performance motivating further research. Another idea is to evaluate in parallel the ODE using batching, and then, collect all sub-results for the final ODE result.
- Other potential ideas is to approximate the Riemannian metric with surrogates leading to simpler ODE systems or providing approximations for the current ODE e.g., considering a diagonal Hessian approximation in the ODE as reviewers BssD and dyBq suggested.

**ODE solver**:
- For solving the ODE system we use a general purpose Python solver (``scipy.integrate.solve_ivp``) that runs on the CPU. When the solver needs to evaluate the ODE, our automatic-differentiation based approach runs on the GPU, and the result is moved to the CPU (``.detach().cpu().numpy()``) causing a significant overhead. Especially when dimensions increase both the transfer of the data and the computations on the CPU are sub-optimal. Implementing an ODE solver on a suitable automatic-differentiation framework (e.g. JAX) solely running on GPU will dramatically improve performance.
- As we know the structure and behavior of our ODE system (geodesics start from low loss which increases along the curve), a potential future work would be to develop solvers that exploit this information. Usually, general purpose solvers aim for accuracy, while in our case even inexact solutions could be potentially useful if computed fast [1].
- The benchmark result (see attached PDF) shows that, in general, increasing data and dimensionality makes the ODE system potentially more expensive to solve. However, there are cases where the solution of the ODE is fast in big models. Related research shows that the loss landscape of overparametrized models exhibits behaviors that may be beneficial to our method e.g., easily connected minima with (nonlinear) continuous paths. Therefore, we believe that even in high dimensions, our method has the potential to scale. It may also inspire new ways to study generalization via the loss landscape.

The cost of evaluating a single ODE is $O(SWN)$, where $S$ is the number of steps of the solver, $W$ is the number of model parameters and $N$ is the dataset size. This is the complexity overhead on top of usual Laplace approximation. However, the number of steps the solver needs to converge mainly depends on the complexity of the associated ODE problem, which is defined by the geometry of the loss landscape and the initial velocity. In the attached PDF we include a benchmark synthetic example that shows how the runtime changes with respect to the size of the model and dataset. While increasing the model parameters and dataset size affect $W$ and $N$, it may be the case that ODE systems do not necessarily get harder, therefore the value of $S$ would be small. In other words, evaluating the ODE becomes more expensive as dimensions increase, but perhaps the actual system gets easier to solve.
With the current implementation we manage to solve ODEs up to ~44000 dimensions (LeNet), which is surprising from the differential geometry perspective, and show that the method performs well in this regime (see attached PDF), but more sophisticated implementations will speed this up. Moreover, in a practical application samples are generated offline and not during test time.

Overall, in this paper we propose a Riemannian extension to Laplace approximation and empirically verify the claims, by considering the complete formulation and relying on pre-existing tools for the implementation. As reviewers acknowledge there is a spectrum of potential research ideas in between for either improving the efficiency of our approach or for new techniques based on the same differential geometric principles. This is in spirit related to NeuralODEs where the original paper proposed the main concept and follow-up works improve parts of it, for example, specialized ODE integrators [2,3,4,5,6].

**References**
1. "Fast and robust shortest paths on manifolds learned from data". G. Arvanitidis et al., AISTATS 2019
2. "Opening the blackbox: accelerating Neural Differential Equations by regularizing internal solver heuristics". A. Pal et al., ICML 2021
3. "On numerical integration in Neural Ordinary Differential Equations". A. Zhu et al., ICML 2022
3. "On robustness of Neural Ordinary Differential Equations". H. Yan et al., arXiv 2020
5. "MALI: A memory efficient and reverse accurate integrator for Neural ODEs". J. Zhuang et al., arXiv 2021
6. "STEER: Simple temporal regularization for Neural ODEs". A. Ghosh et al., arXiv 2020

---

### Decision · Program_Chairs · 2023-09-21

**Decision:**

Accept (poster)

**Comment:**

All reviewers were positive about the methodological contribution as the combination of Laplace methods and Riemannian geometry applying to Bayesian neural networks is interesting and should be of interest to the community. One significant drawback is the high complexity of the proposed method (the run-time required per example is very high) and this explains the small scale of the networks/datasets considered. Would be great if the authors could (partly) address these issues, improve the exposition (see Reviewer dyBq's comment), and add more detailed descriptions to all figures.